# ADAR1 mediated regulation of neural crest derived melanocytes and Schwann cell development

Nadjet Gacem [1,2], Anthula Kavo[2,3], Lisa Zerad[1], Laurence Richard[4], Stephane Mathis [5], Raj P. Kapur[6], Melanie Parisot [7], Jeanne Amiel[1], Sylvie Dufour [2,3], Pierre de la Grange[8], Veronique Pingault[1,9], Jean Michel Vallat[4] & Nadege Bondurand[1,2]*

The neural crest gives rise to numerous cell types, dysfunction of which contributes to many disorders. Here, we report that adenosine deaminase acting on RNA (ADAR1), responsible for adenosine-to-inosine editing of RNA, is required for regulating the development of two neural crest derivatives: melanocytes and Schwann cells. Neural crest specific conditional deletion of *Adar1* in mice leads to global depigmentation and absence of myelin from peripheral nerves, resulting from alterations in melanocyte survival and differentiation of Schwann cells, respectively. Upregulation of interferon stimulated genes precedes these defects, which are associated with the triggering of a signature resembling response to injury in peripheral nerves. Simultaneous extinction of MDA5, a key sensor of unedited RNA, rescues both melanocytes and myelin defects in vitro, suggesting that ADAR1 safeguards neural crest derivatives from aberrant MDA5-mediated interferon production. We thus extend the landscape of ADAR1 function to the fields of neural crest development and disease.

[1] Laboratory of Embryology and Genetics of Human Malformation, Imagine Institute, INSERM UMR 1163, Universite Paris Descartes—Universite de Paris, Paris, France. [2] INSERM, U955, Equipe 06, 8, rue du General Sarrail, 94010 Creteil Cedex, France. [3] Faculte de Medecine, Universite Paris Est, 94000 Creteil, France. [4] Department of Neurology, Centre de Reference Neuropathies Peripheriques Rares, 2 avenue Martin-Luther-King, 87042 Limoges, France. [5] Department of Neurology (Nerve-Muscle Unit) and Grand Sud-Ouest National Reference Center for Neuromuscular Disorders, CHU Bordeaux, Pellegrin Hospital, 33076 Bordeaux, France. [6] Department of Pathology, Seattle Children's Hospital and University of Washington, 4800 Sand Point Way NE, Seattle, WA 98105, USA. [7] Genomics Core Facility, Institut Imagine-Structure Federative de Recherche Necker, INSERM U1163 and INSERM US24/CNRS UMS3633, 24 bvd Montparnasse, 75015 Paris, France. [8] GenoSplice, Paris, France. [9] Service de Genetique Moleculaire, Hopital Necker-Enfants-Malades, 149 rue de Sevres, 75015 Paris, France. *email: nadege.bondurand@inserm.fr

Adenosine-to-inosine (A-to-I) RNA editing is a well-known contributor of transcriptomic and, to a lesser extent, proteomic diversity[1]. Because the resulting inosines are subsequently read as guanosines, this post-transcriptional modification can indeed affect protein function, splicing, RNA stability, transcriptional regulation and miRNA processing[2,3]. A-to-I editing is catalyzed by enzymes of the ADAR family (adenosine deaminase acting on double stranded (ds)RNA; ADAR1-3)[4,5]. Available data suggest that *Adar2* is mainly expressed in brain and is the primary editor of nonrepetitive coding sites[6]. In contrast, ADAR1 has been shown to be the primary editor of repeat elements (Alu and SINES) in noncoding sequences[1].

*Adar1* is widely expressed and is the most highly expressed ADAR outside the central nervous system[1,4,5,7]. In mice, *Adar1* deletion is lethal between embryonic days (E)11.5 and 13.5, due to fetal liver disintegration[8]. Hematopoietic progenitors depend on *Adar1* for their survival and maturation[8,9]. Transcriptional profiling of these cell types has highlighted the activation of interferon (IFN) along with a large number of interferon-stimulated genes (ISGs) upon *Adar1* deletion[10,11]. The embryonic lethality of *Adar1* mutants is rescued upon concomitant deletion of either *Ifih1* or *Mavs*, encoding the cytosolic dsRNA sensor MDA5 (melanoma differentiation-associated protein 5) and MAVS (mitochondrial antiviral-signaling protein), respectively[10–12]. Overall, these data suggest that the key in vivo function of ADAR1 is to mark, via editing, endogenous dsRNAs as self and prevent their immune recognition by MDA5, safeguarding cells from unwanted MDA5-mediated IFN production and ISGs expression in a cell-type-specific manner. However, an MDA5-MAVS-independent function for ADAR1 is described in the development of multiple organs[11].

In humans, mutations of *ADAR1* are responsible for Aicardi-Goutières Syndrome (AGS, OMIM: 615010[13–15]), an inflammatory encephalopathy also referred to as a type 1 interferonopathy, and Dyschromatosis Symmetrica Hereditaria (DSH, OMIM: 127400[16–18]), characterized by hypo- and hyperpigmentation macules on the extremities that appear in infancy. Skin pigmentation defects suggest that ADAR1 is important for melanocytes, at least after birth[18,19].

Melanocytes, specialized skin cells that produce the melanin, are one of the numerous cell types that originate from a transient embryonic structure called neural crest (NC)[20–22]. Upon neural tube closure, NC-derived cells migrate throughout the embryo, and differentiate into melanocytes (mostly found in epidermis and hair follicle)[20,22], but also into cells contributing to septation of the cardiac outflow tract and ventricles, skeletal and connective tissue components of the head, and most neurons and glia of the peripheral nervous system (PNS) including the enteric nervous system and myelin producing-Schwann cells (SCs)[23,24]. Along nerves, SCs precursors develop into immature SCs, then into promyelinating SCs, which establish a one-to-one relationship with large caliber axons, before finally transforming into myelin-forming mature SCs within the first two postnatal weeks in mice[24–26].

The coordinated action of signal transduction cascades, transcription factors and miRNA regulation converge to control the various steps of the development of melanocytes and SCs[22,26–30]. To explore the involvement of RNA editing in the development of NC cells, we generated mouse lines with NC-cell-specific deletion of *Adar1*. The mutants show depigmentation and absence of myelin within the peripheral nerves. Presented analyses suggest that ADAR1 safeguards both melanocytes and SCs from unwanted activation of ISGs and subsequent cell survival and differentiation alterations, respectively. Our results thus extend the landscape of the functional roles of this enzyme and A-to-I RNA editing to the fields of NC development and NC-linked disorders.

## Results

**Depigmentation and early lethality upon NC deletion of *Adar1*.** We studied the requirement of A-to-I RNA editing in NC cells in vivo by breeding mice harboring a floxed (fl) *Adar1* allele with mice expressing Cre recombinase driven by the human tissue plasminogen (HtPA) promoter, triggering ablation of the floxed alleles in NC cells from E9[31]. A YFP tracer (R26R) was also incorporated to specifically mark NC cells and their derivatives[32] (crosses in Fig. 1a and Methods). At birth (P0), the heterozygous as well as the *HtPA-Cre; Adar1^{fl/fl};R26R* mutant animals (referred to as mutants) were present at the expected Mendelian ratio (Fig. 1b) and were not discernable from their wild-type counterparts macroscopically (similar size and weight, Fig. 1c). Four days later (postnatal day 4, P4), the surviving mutants showed a significant reduction in body weight and almost total lack of pigmentation (Fig. 1c). From P4 onwards, mutants additionally developed tremors, were largely unresponsive to stimuli, runted, whereas heterozygotes were still indiscernible from the wild-types. All mutants died within the first 10 days of life (Fig. 1d). Overall, these data suggest alterations of melanocytes and the peripheral nervous system (tremors) upon homozygous *Adar1* deletion.

**Absence of melanocytes in the skin of mutants after birth.** To determine the origin of the absence of pigmentation, we first performed a thorough histological analysis of the skin of wild-type, heterozygous (both referred as controls), and homozygous mutant mice, at P4. No specific alterations were evidenced, apart from the almost total absence of melanin from hair follicles of the mutants (Fig. 2a). RT-qPCR experiments on RNA extracted from dissected skin of controls and mutants revealed downregulation of the expression of melanoblasts markers (such as *Sox10* and the melanocyte-specific isoform of *Mitf*) and melanocyte differentiation genes (*Dct, Trpm1*, and *Tyr*) in the mutants relative to controls (Fig. 2b). Use of the reporter R26R allele under the control of the Cre recombinase, to specifically mark NC cells and their derivatives on skin sections, additionally revealed a reduction of 96% in the percentage of hair follicles containing YFP-positive cells in mutants relative to controls (*HtPA-Cre; Adar1^{fl/+}*) (Fig. 2c). In the very few mutant bulbs still containing YFP-positive cells, the number of these cells was also reduced although they produce melanin (Fig. 2d). Altogether, results suggest that lack of pigmentation upon *Adar1* conditional deletion results from the lack of melanocytes, rather than from the absence of pigment within viable melanocytes.

Based on these observations, we reanalyzed the *Adar1* mutant pups at birth. Although not detectable by gross examination, RT-qPCR experiments (Fig. 3a, first graph) showed a drastic reduction of melanocyte-expressed genes in mutants relative to controls, thus suggesting that the defect is already present.

We next decided to breed the *Adar1* model with mice expressing Cre recombinase driven by the *Wnt1*-promoter (triggering ablation in NC from E8.5[33]). In this model, all the homozygous *Adar1* mutant animals died within their first day of life for unknown reason. Nevertheless, RT-qPCR experiments and use of the YFP tracer showed alterations identical to the ones described above (Supplementary Fig. 1). Our data therefore collectively demonstrate that, irrespective of the Cre driver used, NC-specific deletion of *Adar1* leads to a global lack of pigmentation, resulting from a drastic reduction in the number of melanocytes.

**NC *Adar1* deletion alters survival of melanocytes from E18.5.** To determine the timing and cellular origin of defects observed, we performed similar RT-qPCR experiments at two embryonic

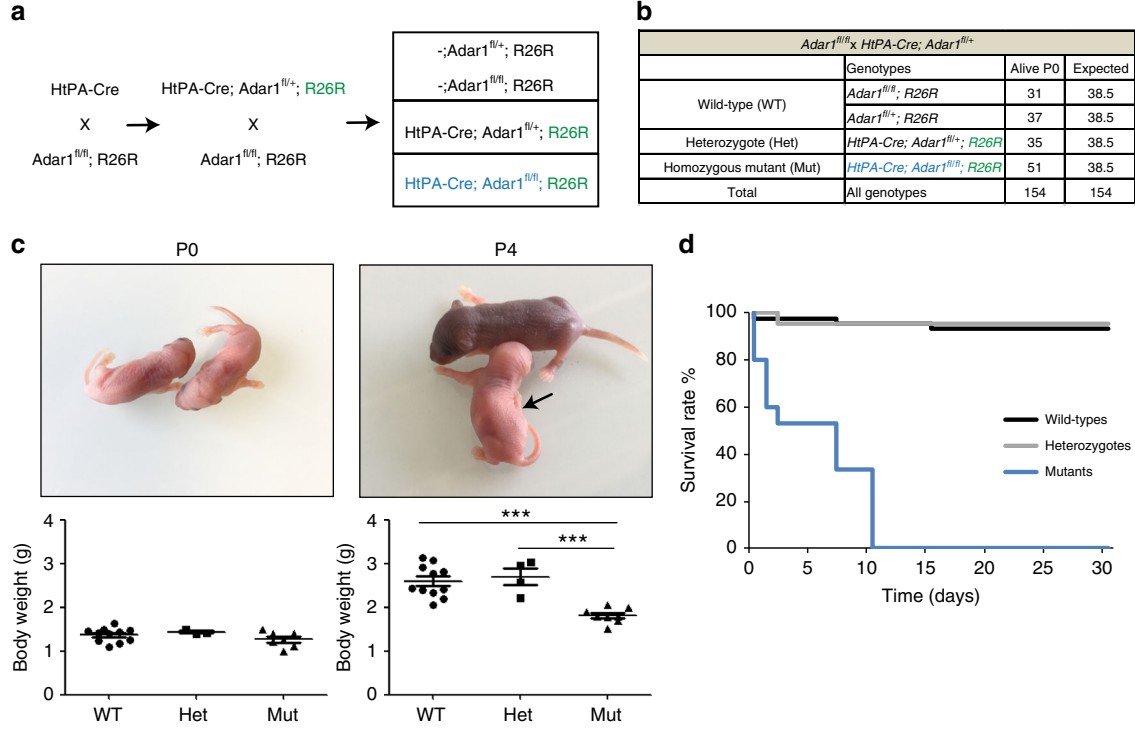

**Fig. 1 NC *Adar1* deletion leads to global depigmentation, tremor, and early lethality in mice. a** Breeding strategy used to generate neural crest-specific *Adar1* conditional knockout mice making use of the HtPA-Cre line. YFP allele is indicated in green; mutant genotype in blue. **b** Genotypes distribution and viability analysis at birth (postnatal day 0, P0). The number of animals of each genotype was compared to the expected ones to analyze viability. **c** Photos and body weight graphs of wild-types (WT, $n = 11$ at each stage), heterozygotes (Het, $n = 3$ at P0 and $n = 4$ at P4) and homozygous mutants (Mut, $n = 7$ at each stage) mice at P0 and P4. Note the significant body weight reduction at P4 and the total depigmentation of mutant mice. Asterisks represent $p$ value: ***$p < 0.001$ determined using one-way ANOVA. **d** Survival rate (%) of wild-types (black), heterozygotes (gray) and mutants (blue) over time (days). Note that all mutants died within the first 10 days of life.

stages (E16.5 and 18.5) and at birth to compare the expression of three melanocytic markers (*Sox10*, *Dct, Tyr*) (Fig. 3a–d; first column), and six ISGs (*Cxcl10, Isg15, Ifit1, Ifit2, Rsad2*, and *Mx1*, Fig. 3a–d second column), a signature recently shown to be increased upon *Adar1* deletion in other tissues[11]. As previously mentioned, all the mutants showed marked downregulation of the expression of the three melanocytic genes tested at birth. This was concomitant to an elevated ISG signature relative to that of control littermates (Fig. 3a). At E18.5, half of the mutants showed downregulation of *Dct* and *Tyr* and strong upregulation of the ISG signature relative to controls (Fig. 3b). The others showed normal expression of *Sox10, Dct*, and *Tyr*, and a slight increase of ISGs (Fig. 3c). In contrast, there was no difference between the mutants and controls at E16.5 (Fig. 3d), a stage at which melanoblasts normally enter the developing hair follicle and start to differentiate[22]. Overall, these results suggest that the migration and differentiation of melanoblasts is not affected by *Adar1* deletion up to E16.5. The disappearance of melanocyte-expressed genes occurs at approximately E18.5 and is preceded by upregulation of the ISG signature, leading us to analyze melanocytes survival at this embryonic stage.

Immunohistochemistry experiments using GFP and activated-caspase 3 antibodies revealed a high number of apoptotic YFP-positive cells in the epidermis and, to a lesser extent, in the hair follicles in the skin of mutants compared to controls at E18.5 (stainings and quantification, Fig. 3e), suggesting that the global depigmentation observed is due to the rapid loss of melanocytes by cell death. Of note, the number of CD45-positive cells was similar in the skin of mutants and controls at E18.5 and P4, suggesting that activation of ISGs and apoptosis occurs without immune cell recruitment (Supplementary Fig. 2).

**NC *Adar1* deletion alters SCs differentiation from birth.** Based on the unsteady gate and tremors observed in the mutant animals, we speculated that *Adar1* deletion may lead to the alteration of SCs development and/or myelin formation[24]. The sciatic nerves of mutants and controls were therefore analyzed, first at P4, by optical and electron microscopy (Fig. 4a–c). At the ultrastructural level, the control SCs had established a 1:1 relationship with axons and begun wrapping a myelin membrane along the axons, whereas the mutant ones failed to initiate myelination (Fig. 4b). Indeed, the axons of mutants were surrounded by cytoplasmic extensions of the SCs, showing no radial sorting alterations; but the SCs stalled at this promyelinating stage (no mesaxons, Fig. 4c). Quantifications confirmed a drastic decrease in the number of myelinated axon profiles per nerve section in mutants compared to controls, while the number of SCs and the axon diameters were similar (Fig. 4d). Both controls and mutants had Remak bundles, suggesting unaffected nonmyelinating SCs. Use of CD45 and (F4/80+CD11b+CD68) antibodies confirmed no, or rather limited, immune cells recruitment (Supplementary Fig. 3). Overall, these data suggest that *Adar1* mutant SCs stall at the promyelinating stage, without evidence of cell degeneration.

These observations were consistent with immunohistochemistry experiments (Fig. 4e), which showed an apparently normal number of YFP-positive cells, an almost complete absence of MBP staining, but very few activated-Caspase3-positive cells in the sciatic nerves of mutants relative to controls (although statistically significant, percentage of apoptotic cells did not exceed 8% in controls and mutants, Fig. 4e). RT-qPCR experiments, performed on RNA extracted from isolated sciatic nerves, confirmed the reduced expression of several late SCs differentiation markers, such as *Dusp15, Pmp22, Mpz*, and *Mbp* in

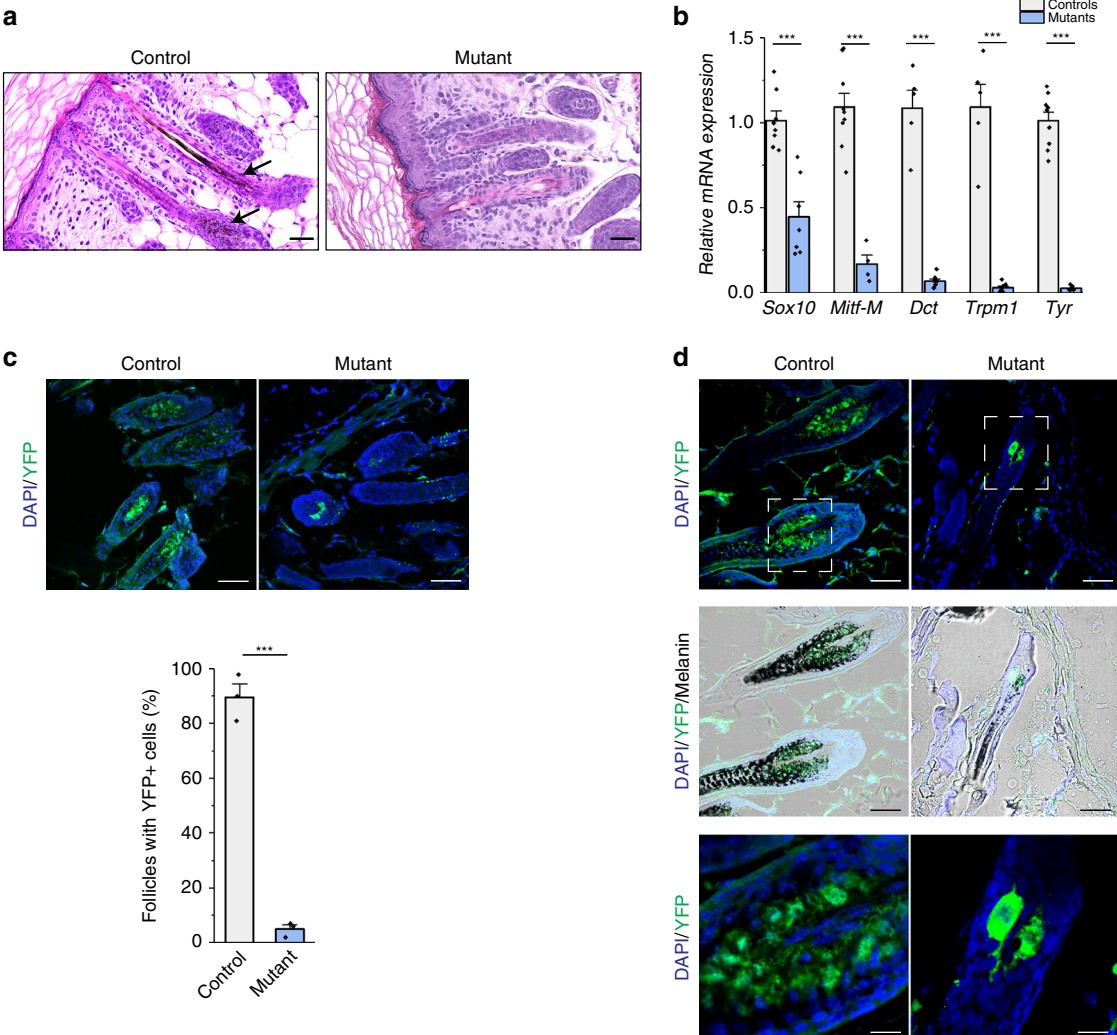

**Fig. 2 NC *Adar1* deletion leads to reduced number of melanocytes in the skin of mutants at P4. a** Paraffin skin sections from control and *HtPA-Cre; Adar1^fl/fl* mutant mice stained with hematoxylin and eosin (H/E). Black arrowheads indicate presence of melanin in controls. Scale bar: 25 μm. **b** Relative expression level of melanocyte-expressed genes in skin of control (gray) and mutant (blue) mice (*n* = 5 or 8 controls and *n* = 4, 7 or 9 mutants depending on tested genes) analyzed by RT-qPCR. The relative abundance values of each amplicon were normalized to the internal control β-actin, and expression level of mutants and controls represented relative to controls. All data represent mean ± SEM, asterisks represent *p* value: ****p* < 0.001 determined using *t* test. **c** Number (%) of follicles containing YFP-positive cells in the skin of control and mutant mice. Representative regions showing YFP (green) cells in follicles and quantification are presented in controls (gray) and mutants (blue). Statistical differences between the groups (*n* = 3 independent controls and mutants) were determined using *t* test (asterisks represent *p* value: ****p* < 0.001). Scale bar: 50 μm. **d** Morphology and functionality of remaining YFP-positive cells (melanocytes) in follicles of control and mutant mice. Melanin synthesis (black) is shown on white light images. Bottom panels represent higher magnifications of upper panels (Scale bars: 50 and 15 μm, respectively). In (**c**) and (**d**) counterstaining with DAPI (blue) is shown to identify structures. Careful analysis of the few remaining YFP-positive cells in mutants skin showed them to be enlarged and able to produce melanin, suggesting that when still present, mutant melanocytes can retain at least part of their functionality. For (**b**) and (**c**), source data are provided as a source data file.

all mutants relative to controls. In contrast, the expression of earlier markers (*Sox10* and *Egr2*) was not altered, confirming that the early development of these cells is not markedly altered by *Adar1* deletion (Fig. 4f).

We next performed similar analyses on older mice (up to P10, latest stage available for *Adar1* mutants) to determine whether *Adar1* deficiency leads to a delay in or blockage of SCs differentiation. Semioptical and electron microscopy showed very little myelin in the mutants at P8 and P10 relative to controls (Supplementary Fig. 4a, b). Quantification again confirmed the drastic decrease in the number of myelinated axon profiles per nerve section, but no change in axon diameter at both stages (Supplementary Fig. 4c), suggesting a blockade of differentiation rather than a simple delay.

In parallel, we determined the onset of defects. Analysis of the expression of three SCs markers (*Pmp22, Mpz* and *Mbp*) and the ISGs signature revealed significantly greater ISGs expression in all mutants than controls from E18.5 (Supplementary Fig. 5b). This was followed by downregulation of the expression of the SCs markers from birth (Supplementary Fig. 5a), suggesting that SCs differentiation is blocked around birth and preceded by the upregulation of ISGs signature.

As with pigmentation, we confirmed these results using the *Wnt1-Cre*-mediated conditional deletion mouse model. At birth, electron microscopy showed a few myelinating SCs in controls but none in the mutants (Supplementary Fig. 6a). RT-qPCR experiments confirmed lower expression of *Dusp15, Pmp22, Mpz,* and *Mbp* in the mutants, but similar expression of *Sox10* and *Egr2*

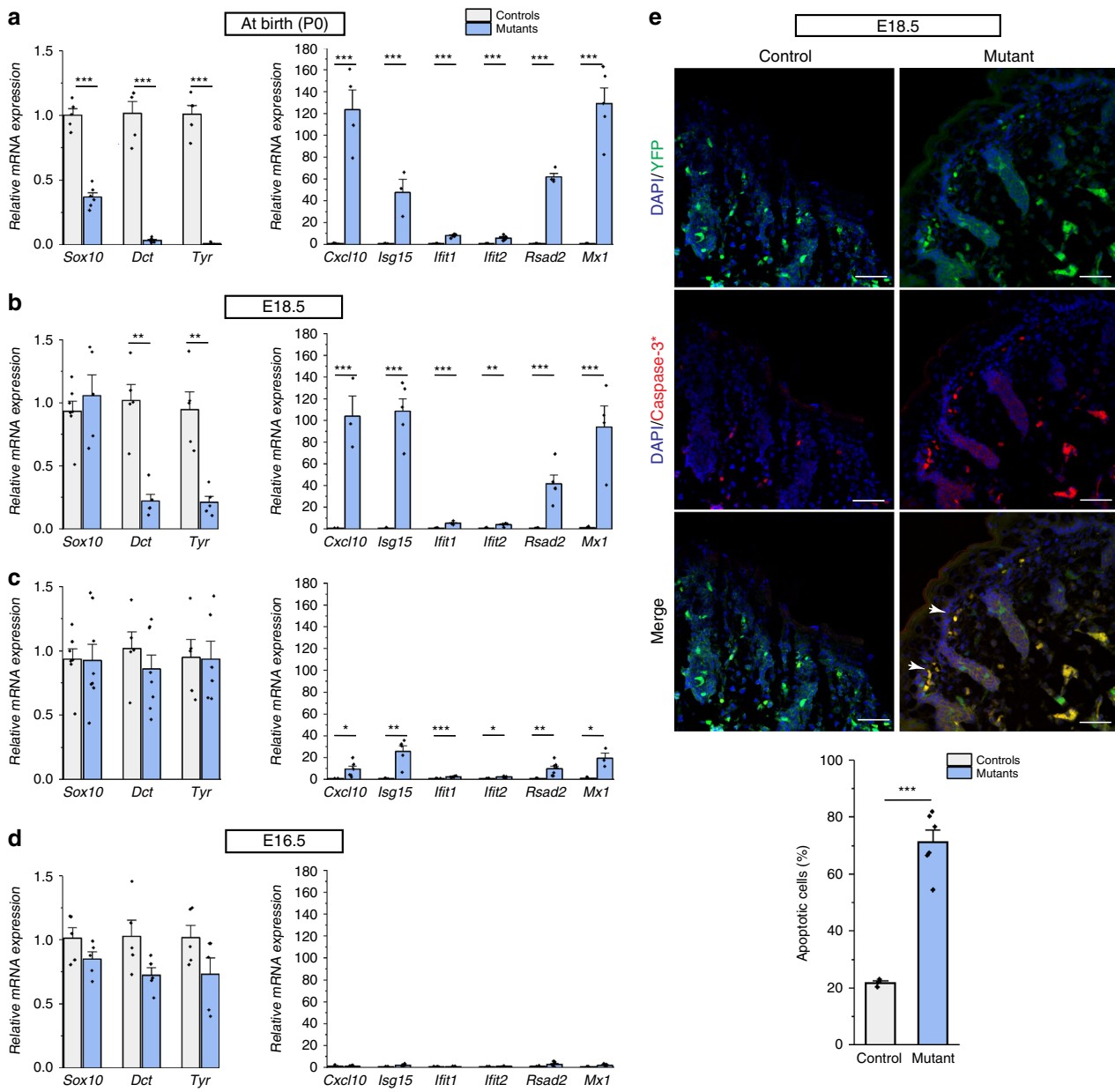

**Fig. 3 ADAR1 activity is required to control the survival of melanocytes from E18.5. a–d** Analysis of the expression of melanocytes differentiation genes (*Sox10, Dct, Tyr*) and Interferon-stimulated genes (ISG signature composed of *Cxcl10, Isg15, Ifit1, Ifit2, Rsad2, Mx1*) in the skin of controls (gray) and *HtPA-Cre; Adar1*fl/fl mutants (blue) at **a** P0, **b**, **c** E18.5 (note that we found mutant embryos of the two categories within the same litter) and **d** E16.5, by RT-qPCR. All data represent mean ± SEM. Statistical differences between the groups (for **a** *n* = 4 controls and *n* = 3–6 mutants depending on tested genes; for **b** *n* = 5 controls and *n* = 4 or 5 mutants depending on tested genes; for **c** *n* = 4 or 5 controls and *n* = 4–8 mutants depending on tested genes; for **d** *n* = 5 or 8 controls and *n* = 5 or 8 mutants depending on tested genes) were determined using *t* test (asterisks represent *p* values: *\**p* < 0.05, \*\**p* < 0.01, \*\*\**p* < 0.001). **e** YFP (green) and activated-caspase 3 (red) immunostaining on skin sections of control and *HtPA-Cre; Adar1*fl/fl mutant mice at E18.5 (white arrowheads indicate apoptotic cells) and quantification of caspase-3-positive cells among YFP cells (%). Data represent mean ± SEM, and statistical differences between the groups (*n* = 3 controls, gray and *n* = 6 mutants, blue) was determined using *t* test (asterisks represent *p* value: \*\*\**p* < 0.001). Counterstaining with DAPI (blue) is shown to identify structures. Scale bar: 50 μm. For all panels, source data are provided as a source data file.

(Supplementary Fig. 6b). The early development of SCs is thus not altered by *Adar1* deletion in either mouse line, but SCs appears to stall at or near promyelinating stage.

**NC *Adar1* deletion does not affect all NC derivatives equally.** We determined if other NC derivatives were affected by *Adar1* deletion. Analysis of three of these: enteric nervous system, heart structures derived from NC cells and craniofacial derivatives was

performed. Macroscopic, histopathological analyses and Alcian blue followed by Alizarin red coloration revealed no obvious alterations of these three NC derivatives in mutants compared to controls at P4 with the techniques used (Supplementary Fig. 7a–c). However, 30% of the mutants (4 out of 13) present with cleft palate, suggesting some craniofacial alterations might be an incompletely penetrant feature.

RT-qPCR experiments on each tissue revealed increased ISGs expression in mutants relative to controls, but no alteration of

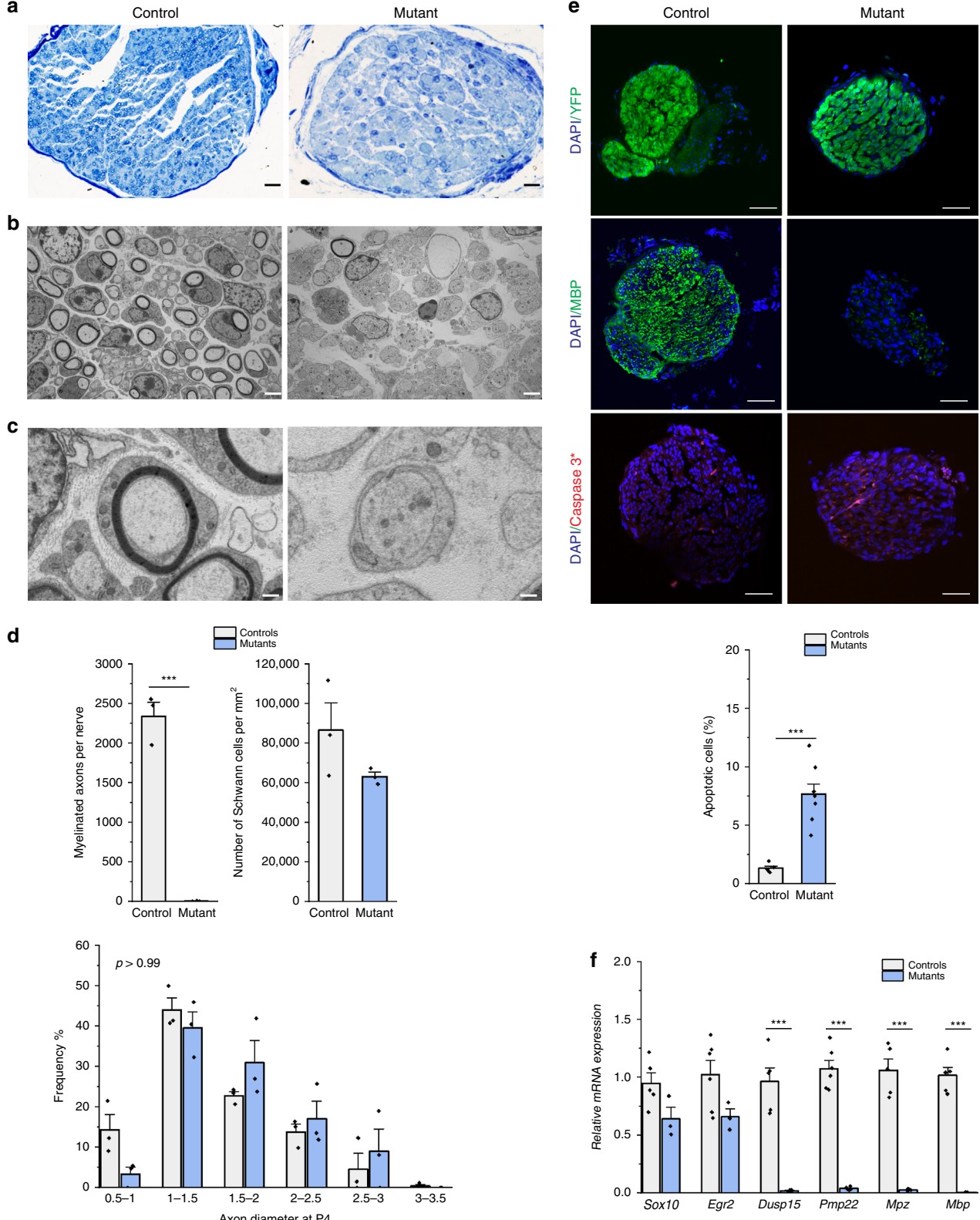

NC expressed genes tested at P4 (Supplementary Fig. 7a–c). ISGs expression was however not increased in the liver of mutants (Supplementary Fig. 8), suggesting that this response is not systemic. Although we cannot exclude functional or cell-type-specific alterations not tested here, our results suggest *Adar1* deletion does not affect all NC derivatives equally.

**RNA-seq analysis in sciatic nerves of controls and mutants**. To further define origin of *Adar1* mutant SCs arrest and assess global changes in gene regulation, we performed transcriptomic analysis (RNA-seq) of sciatic nerves isolated from three control and three *HtPA-Cre; Adar1$^{fl/fl}$;R26R* mutant animals at P4. The steady-state level of 3009 mRNAs differed by at least twofold between the two

**Fig. 4 ADAR1 activity is required to control myelin formation within sciatic nerves at P4. a** Toluidine blue-staining and **b**, **c** electron micrographs of transverse sections of sciatic nerves from control and *HtPA-Cre; Adar1*$^{fl/fl}$ mutant mice, illustrating the almost total absence of myelin in the mutants compared to controls. In mutants, the axons were surrounded by cytoplasmic extensions of the SCs, but the latter stalled at this promyelinating stage (**b** and insets shown in **c**). Note that a minority of mutant SCs myelinate axons properly (see **b**) along with absence of sign of axonal degeneration or B- or T-cell or macrophage invasion in the mutants compared to controls. Scale bars: 10 μm (**a**), 2 μm (**b**), 500 nm (**c**). **d** Quantifications performed from TEM pictures to count myelinated axons number per nerve, the SCs number per square millimeter and the distribution of axon diameter in controls (gray) and *HtPA-Cre; Adar1*$^{fl/fl}$ mutants (blue). Data of the three graphs represent mean ± SEM of $n = 3$ mice/group (Statistical differences between the groups were determined using $t$ test (asterisks represent $p$ value: ***$p < 0.001$, or Kolmogorov–Smirnov test, $p$ values indicated on the graph). **e** DAPI (blue)/YFP (green), DAPI (blue)/MBP(green) or DAPI (blue)/Activated-Caspase 3 (red) immunostaining performed on sections of sciatic nerves from control and *HtPA-Cre; Adar1*$^{fl/fl}$ mutant mice along with the quantification of % of caspase-3-positive cells among YFP-positive cells. Data represent mean ± SEM. Statistical differences between the groups ($n = 5$ independent sections of four controls and $n = 8$ independent sections of five mutants) was determined using $t$ test (asterisks represent $p$ value: ***$p < 0.001$). Scale bar: 50 μm. **f** Relative expression of various SCs markers in sciatic nerve from control (gray) and *HtPA-Cre; Adar1*$^{fl/fl}$ mutant (blue) mice analyzed by RT-qPCR. All data represent mean ± SEM ($n = 5$ or 6 controls and $n = 3$ mutants depending on tested genes). Statistical differences between the groups were determined using $t$ test (asterisks represent $p$ value: ***$p < 0.001$). For (**d**–**f**), source data are provided as a Source data file.

genotypes (Fig. 5a). We confirmed the results for a subset of these genes by RT-qPCR (Fig. 5b). The full list and the top 50-up and 50-downregulated genes are shown in Supplementary data 1 and Fig. 5c. Gene Ontology (GO) term and pathway analyses confirmed the profound enrichment of functions/pathways activated upon inflammation or after viral infection in the mutants versus controls (Fig. 5d, e). Screening of the 3009 differentially expressed RNAs against the interferome database[34] showed that 2253 (75%) are known to be regulated by IFN type 1 or 2 (Fig. 5f). Aside from activation of the innate immune response, the GO terms analysis also showed activation of other biological functions in mutants relative to controls, including growth-factor activity and positive regulation of the epithelial to mesenchymal transition (EMT) (Fig. 5d). The latter being critical for NC development, we looked at expression of key EMT-related genes in our dataset. A decrease in *Cdh1* expression (E-cadherin), and upregulation of the master regulators *Twist1* and *2*, *Snail2*, *Lef1*, and *Met* were observed (Supplementary data 1). However, the expression of well-known mesenchymal markers *Cdh2* (N-cadherin) and *Vimentin* were not altered, suggesting that only part of the molecular machinery associated to EMT is activated upon *Adar1* deletion.

Consistent with myelination alterations, genes encoding myelin proteins were among the top 50-downregulated genes in the *Adar1* mutants relative to controls (Fig. 5c and Supplementary data 1). Additionally, cholesterol biosynthesis and metabolism (one of the main components of myelin sheaths) and lipid metabolism were among the processes significantly enriched in controls relative to the mutants (Fig. 5d, e).

Based on the alterations of SCs observed (block at promyelinating stage), we next analyzed the differential expression of key genes (transcription factors in particular) known to control this specific step of development[26,30]. Among these, expression of *Sox10*, *Egr2*, *Egr3*, *yy1*, *Sox2*, *Jun*, *Id2*, *Pax3*, and *Nfkb1* was not significantly altered in the *Adar1* mutants relative to controls. Of note, the expression of *Egr1*, which is normally downregulated at the promyelinating stage[35], was significantly higher in the *Adar1* mutants (12.5-fold increase, Supplementary data 1). In contrast, the expression of *Pou3f1* and *Pou3f2*, known to be transiently expressed in SCs[26,36] and peaking at the promyelinating stage, were both lower (ratios of 0.23 and 0.44, respectively). The deregulation of the expression of these three candidate transcription factors may be a direct consequence of, contribute to, or be responsible for the observed promyelination blockade of differentiation.

To determine whether additional factors involved in transcriptional regulation are also deregulated, we examined the relative expression of other genes associated with transcriptional regulation (see Methods for the GO term used). Among these, 69 were deregulated more than fivefold in mutants relative to controls. Most of them were upregulated in the mutants (64/69) and 87% were found within the interferome database (Fig. 5g). Among them, we again noticed the presence of *Egr1*. There was also increased expression of *Hes1*, *Myc*, *Tfap2a*, and *Tfap2b*, previously shown to be negative regulators of myelination or transcription factors known for their function in early NC development[26,37]. In addition to *Hes1*, the Notch effector *Hey2* was also upregulated in mutants (4.7-fold-change, Supplementary Data 1), arguing for persistently activated inhibitory Notch signaling in *Adar1* mutants relative to controls. Overall, data suggest that SCs developmental arrest in *Adar1* mutants could be due to the persistent expression of limited number of these regulators of myelination.

**Injury-associated genes activation in *Adar1* mutants**. Intriguingly, the top 50 upregulated genes included several known to be activated upon nerve injury[25,38]. In particular, *Shh* and *Gdnf*, two markers that are not expressed in SCs in normal mature or developing nerves, but known to be activated SCs during repair, were markedly upregulated in the mutant mice (Fig. 5c). We therefore evaluated the extent of the overlap between the two conditions by comparing our RNA-seq data to several lists of mRNAs previously shown to be deregulated 1, 5, or 7 days after nerve injury[39,40]. Strikingly, 34% of the mRNAs deregulated upon *Adar1* deletion are also deregulated upon nerve injury, including decreased expression of myelin genes and those for cholesterol biosynthesis, increased expression of a subset of early SC markers, activation of genes involved in the innate immune response, and increased expression of specific repair markers, such as *Shh*, *Gdnf*, *Artemin*, *Fgf5*, *Wif1*, and *Lgals3*[25,41,42] (Fig. 6a and Supplementary data 2 for the 1028 overlapping mRNAs). The differential expression of some of these was further validated by RT-qPCR (Fig. 6b). Because the morphological transition from differentiated/mature SCs into a repair SCs was also shown to be associated with an EMT signature[39], we compared the datasets. Of the 111 EMT-associated genes found to be differentially expressed between uncut and cut nerves[39], 36% were also deregulated upon *Adar1* deletion (Supplementary Table 1 and Fig. 6c). Overall, analysis of the RNA-seq dataset suggests that activation of a program resembling response to nerve injury occurs upon conditional NC deletion of *Adar1*.

**A-to-I RNAs targets and *Mda5* shutdown rescue experiments**. We next wished to directly determine the contribution of A-to-I editing. Making use of the pipeline described in Methods, we identified 60 A-to-I (G) editing sites specifically altered in

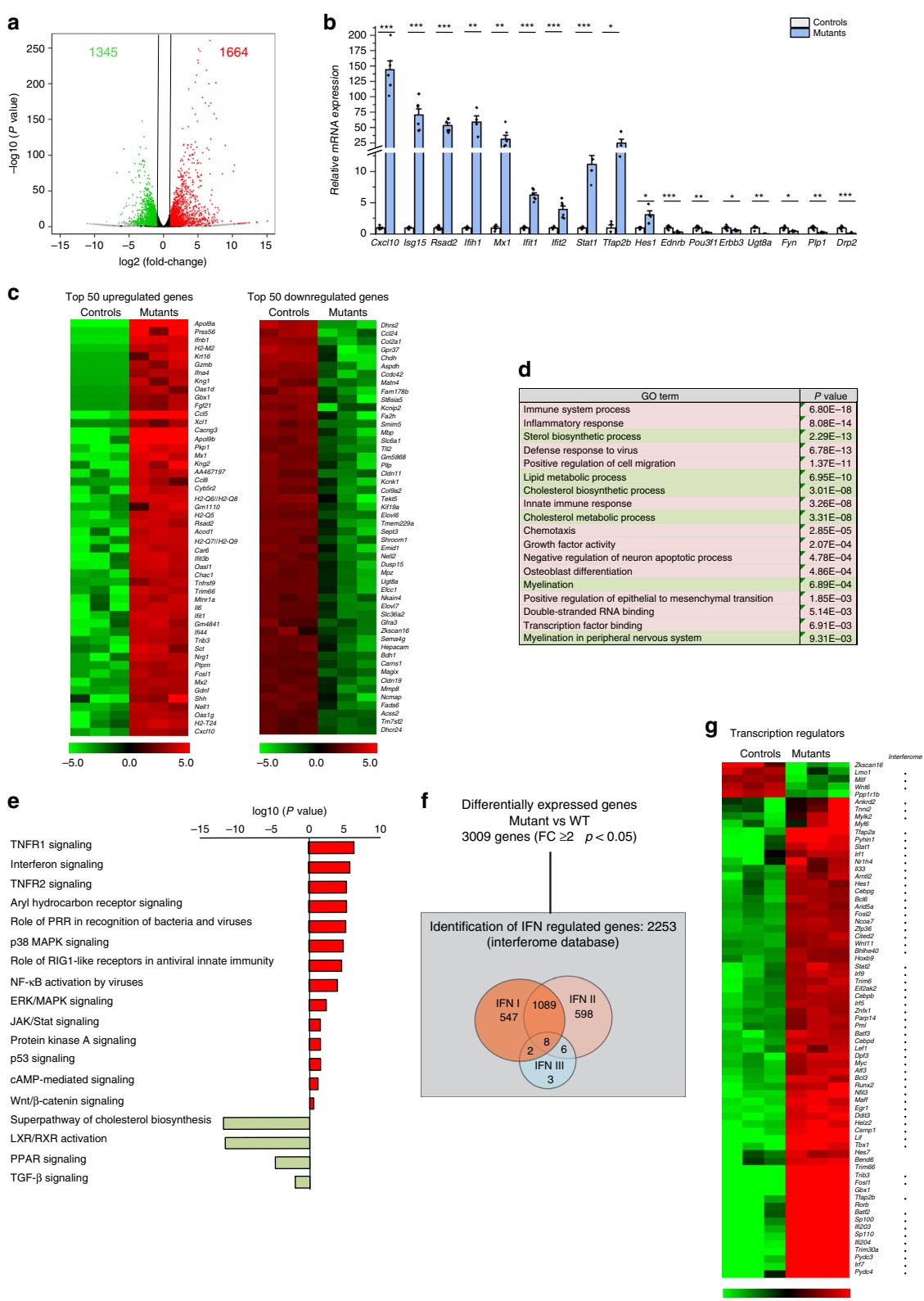

mutants within or near 57 different genes in our RNA-seq dataset (Supplementary Data 3). Only six of these sites were previously reported in the RADAR or DARNED databases, suggesting that most of the identified sites may be regulated in a tissue-specific manner. RNA editing sites led to amino-acid change, intronic, splice acceptor site or 3′UTR variants with similar frequency. Only seven of the 57 targets showed deregulation of over twofold.

All seven targets were in the interferome database, while only three were previously associated to repair: *Atp8b1*, *Gusb*, and *Slc7a11*. We also examined the known function of each of the 57 associated genes. Some (*Ufc1* and *Ctf3c3*) are involved in disorders with CNS myelin alterations (OMIM 618076). NC deletion of *Ilk* leads to alteration of SCs myelination[43]. Nevertheless, expression and splicing of these genes were similar in the mutants

**Fig. 5 Transcriptomic analysis of sciatic nerves in mutants relative to controls. a** Volcano plot of differentially expressed genes in sciatic nerves of *HtPA-Cre; Adar1^{fl/fl}* mutants compared to controls (upregulated in mutants: red; downregulated in mutants: green; unchanged in black). Results were considered statistically significant for *p* values ≤ 0.05 and fold-changes ≥ 2.0 (*n* = 3 for each condition) using Deseq2 R package for differential analyses. In all, 1664 were upregulated in the mutant versus control animals and 1345 were downregulated, suggesting ADAR1 is involved in mechanism governing gene repression and activation. **b** Validation of a subset of differentially regulated genes on RNA extracted from isolated sciatic nerves from controls and mutants by RT-qPCR. Relative expression levels in controls (gray) and mutants (blue) are presented as mean ± SEM. Statistical differences between the groups (*n* = 4 controls and *n* = 4 mutants) were determined using *t* test (asterisks represent *p* values: \**p* < 0.05, \*\**p* < 0.01, \*\*\**p* < 0.001). Source data are provided as a source data file. **c** Heatmap representation of RNA-seq analysis depicting the top 50-up and 50-downregulated genes in mutants relative to controls. Colors reflect the *z*-score. **d** Representation of enriched functions by GO term and enriched signaling pathways (using GO analysis and Ingenuity). **e** Functions/signaling pathways **e**nriched in mutants versus controls are indicated in red, those enriched in controls versus mutants are indicated in light green. **f** The interferome database V2.01l [http://interferome.its.monash.edu.au/interferome/] was used to determine the number of interferon regulated genes (IRGs) specific to different types of Interferon IFN (I, II and III) within the 3009 differentially regulated genes upon *Adar1* deletion. Overlap of 75% (2253/3009) was evidenced. Fisher's exact test confirmed enrichment (*p* < 2.2E-16). **g** Heatmap representation of differentially expressed transcriptional regulators (found in association with GO terms: DNA-templated transcription, positive or negative regulation of transcription from RNA polymerase II promoter, positive or negative regulation of transcription, DNA-templated transcription factor activity, sequence-specific DNA binding, and positive regulation of gene expression). Out of the 281 factors identified, 69 were deregulated more than fivefold in mutants relative to controls. Name of each gene is indicated on the left, along with its belonging to the Interferome dataset. Colors reflect the *z*-score.

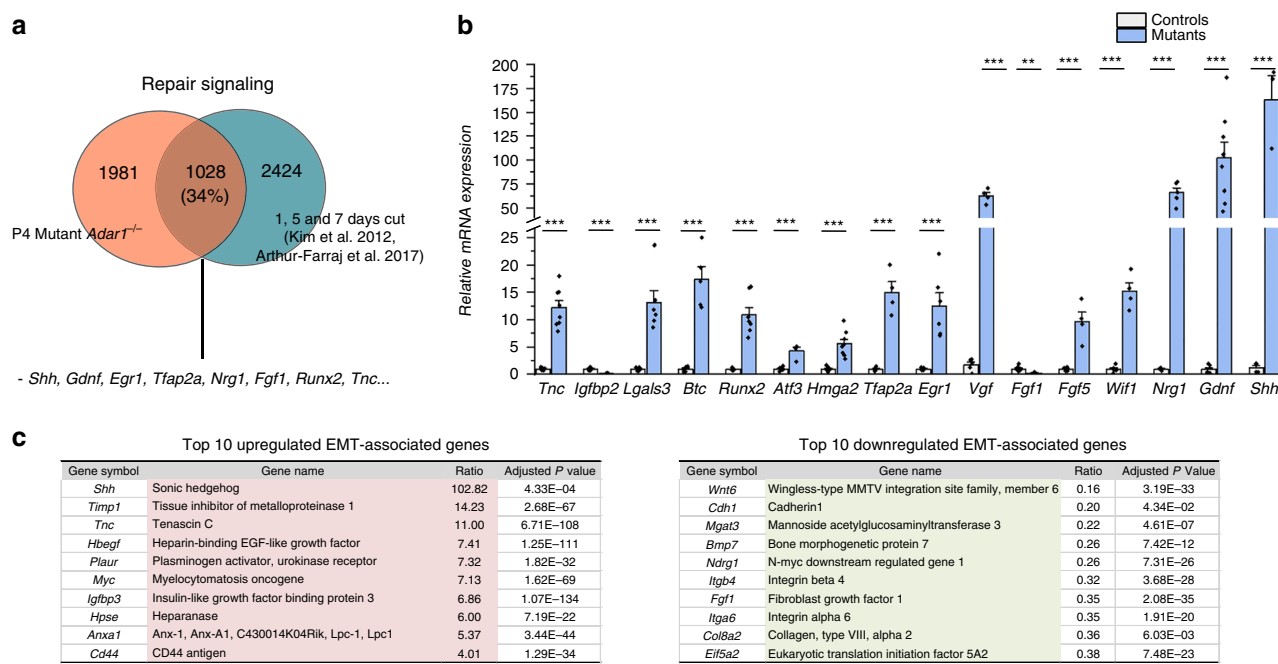

**c**

Top 10 upregulated EMT-associated genes

| Gene symbol | Gene name | Ratio | Adjusted *P* value |
|---|---|---|---|
| *Shh* | Sonic hedgehog | 102.82 | 4.33E-04 |
| *Timp1* | Tissue inhibitor of metalloproteinase 1 | 14.23 | 2.68E-67 |
| *Tnc* | Tenascin C | 11.00 | 6.71E-108 |
| *Hbegf* | Heparin-binding EGF-like growth factor | 7.41 | 1.25E-111 |
| *Plaur* | Plasminogen activator, urokinase receptor | 7.32 | 1.82E-32 |
| *Myc* | Myelocytomatosis oncogene | 7.13 | 1.62E-69 |
| *Igfbp3* | Insulin-like growth factor binding protein 3 | 6.86 | 1.07E-134 |
| *Hpse* | Heparanase | 6.00 | 7.19E-22 |
| *Anxa1* | Anx-1, Anx-A1, C430014K04Rik, Lpc-1, Lpc1 | 5.37 | 3.44E-44 |
| *Cd44* | CD44 antigen | 4.01 | 1.29E-34 |

Top 10 downregulated EMT-associated genes

| Gene symbol | Gene name | Ratio | Adjusted *P* Value |
|---|---|---|---|
| *Wnt6* | Wingless-type MMTV integration site family, member 6 | 0.16 | 3.19E-33 |
| *Cdh1* | Cadherin1 | 0.20 | 4.34E-02 |
| *Mgat3* | Mannoside acetylglucosaminyltransferase 3 | 0.22 | 4.61E-07 |
| *Bmp7* | Bone morphogenetic protein 7 | 0.26 | 7.42E-12 |
| *Ndrg1* | N-myc downstream regulated gene 1 | 0.26 | 7.31E-26 |
| *Itgb4* | Integrin beta 4 | 0.32 | 3.68E-28 |
| *Fgf1* | Fibroblast growth factor 1 | 0.35 | 2.08E-35 |
| *Itga6* | Integrin alpha 6 | 0.35 | 1.91E-20 |
| *Col8a2* | Collagen, type VIII, alpha 2 | 0.36 | 6.03E-03 |
| *Eif5a2* | Eukaryotic translation initiation factor 5A2 | 0.38 | 7.48E-23 |

**Fig. 6 Activation of an injury-associated gene expression program in mutants. a** Venn diagram showing significant overlap between RNA-seq data generated in this study (orange) and differentially expressed genes upon peripheral nerve injury (blue)[39,40]. The number of genes in each category, the % of overlap and representative genes are indicated. Fisher's exact test confirmed enrichment (*p* < 2.2E-16). Note that 1028 out of 3009 genes we found deregulated are overlapping with published datasets. **b** Validation of a subset of differentially expressed genes by RT-qPCR. All data represent mean ± SEM. Statistical differences between the groups (*n* = 4–6 controls and *n* = 4–8 mutants depending on tested genes) were determined using *t* test (asterisks represent *p* values: \**p* < 0.05, \*\**p* < 0.01, \*\*\**p* < 0.001). Source data are provided as a source data file. **c** Of the 111 EMT-associated genes found to be differentially expressed between uncut and cut nerves[39], 34% were also deregulated upon *Adar1* deletion. Top 10 up-(pink) and down-(green) regulated genes found in common, (*p*-adj < 0.05) are indicated. Note that "EMT-associated genes" are more generally associated to migration and cell adhesion function.

and the controls upon bioinformatics analysis. *Slc7a11* is the only gene found to be deregulated and with a probable association with myelin production.

Irrespective of the identity of the targeted RNA, it seemed likely that alterations of A-to-I mismatches could trigger chronic activation of ISGs after their recognition by MDA5 as in other tissues[10]. We therefore tested whether concomitant deletion of *Ifih1* (encoding Mda5) could rescue the SCs defects observed. In primary cultures of mixed neurons and SCs under enhanced myelinating conditions[44], we first observed that MBP-positive segments (mark of myelination) were absent from the neurons (labeled by TUJ1) of mutants while present in controls (Fig. 7a).

As in vivo, these defects were accompanied by upregulation of the expression of the two tested ISGs relative to controls (*Cxcl10* and *Isg15*, Fig. 7b). Strikingly, we observed MBP-positive myelin segments in *Adar1* mutants transduced with Sh-*Ifih1* (>60 segments per cultures), but none in those transduced with the Sh-control (Fig. 7a). This rescue was concomitant with a return of the two ISGs expression to basal levels (Fig. 7b). Altogether, our results show that *Adar1* safeguards SCs from unwanted MDA5-mediated ISGs activation, a process that leads to the alteration of myelin formation if deregulated.

On the same line, we tested whether concomitant deletion of *Ifih1* could rescue the melanocytes defects observed. Trunk neural

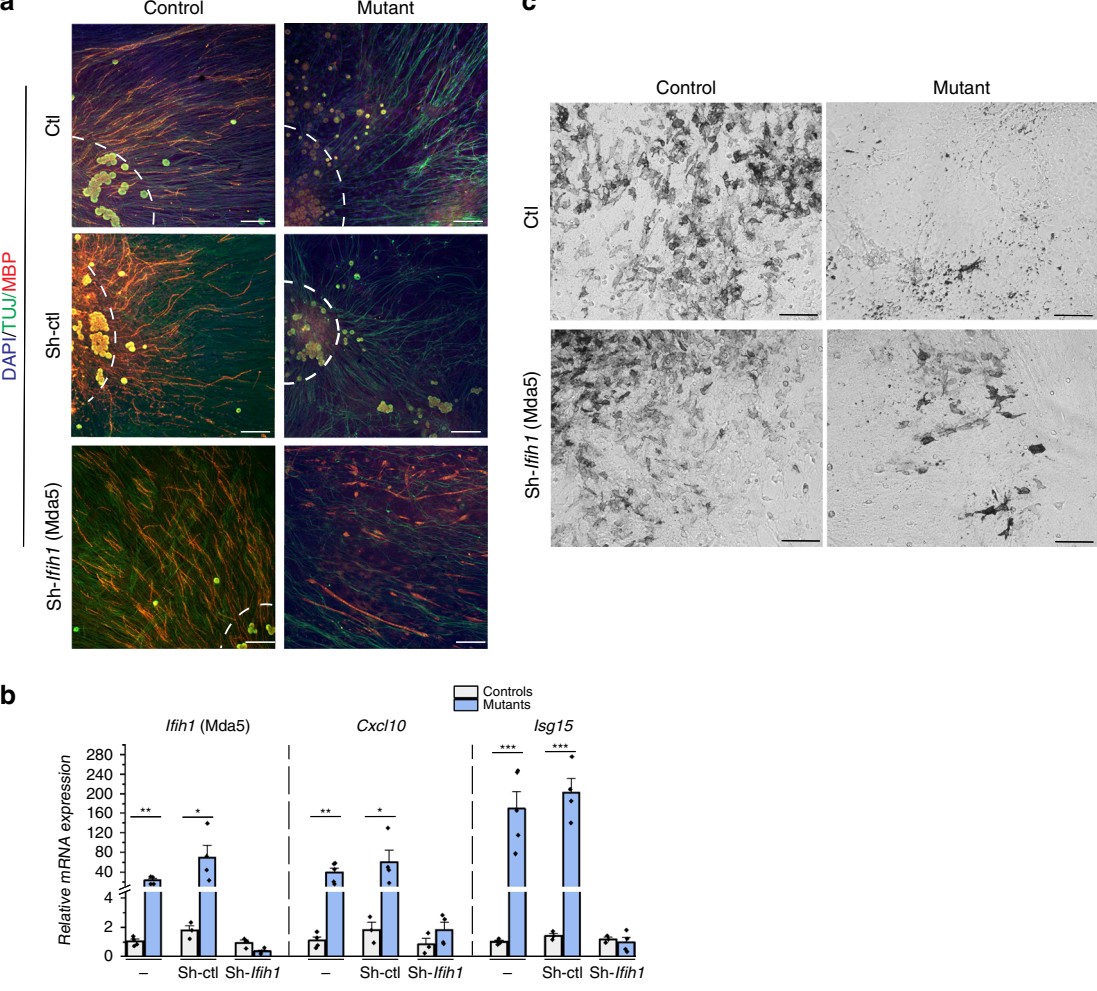

**Fig. 7 Shutdown of *Ifih1* rescues the SCs and melanocytes defects in *Adar1* mutants. a** TUJ1 (green)/MBP (red) immunostaining performed on primary cultures of mixed neurons and SCs from dorsal root ganglia (DRG) of control and *HtPA-Cre; Adar1fl/fl* mutant embryos at E13.5. Cultures were untransduced (no infection, Ctl) or transduced with lentivirus containing Sh-control (Sh-ctl) or Sh-*Ifih1* just before myelination induction. Note the absence of MBP in *Adar1* mutants and the restored phenotype upon Sh-*Ifih1* transduction. Counterstaining with DAPI (blue) is shown to identify cell density. Scale bar: 12 μm. **b** Relative mRNA expression of *Ifih1* (Mda5, to show *Ifih1*-shRNAs efficacy validation), and two ISGs (*Cxcl10 and Isg15*) analyzed by RT-qPCR on RNAs extracted from untransduced (−) and transduced primary cultures of controls (gray) and mutants (blue). All data represent mean ± SEM. Statistical differences between the groups ($n = 4$ controls and $n = 4$ mutants) were determined using $t$ test (asterisks represent $p$ values: *$p < 0.05$, **$p < 0.01$, ***$p < 0.001$). **c** Pigmented melanocytes (melanin-producing cells) observed after culture of NC cells issued from trunk region of neural tube of control and *Adar1* mutant embryos at E9.5. No discernable differences could be observed between controls and mutants during the first 7 days of culture, but note the reduction of melanin-producing cells in *Adar1* mutants ($n = 7$) compared to controls ($n = 6$) after 21 days of culture. Three days after neural tube explant deposition on fibronectin, cultures were untransduced (Ctl) or transduced with lentivirus containing Sh-*Ifih1*. Note the restored phenotype of mutants upon Sh-*Ifih1* transduction ($n = 5$). Sh-*Ifih1* transduction had no discernable effects on control cultures ($n = 3$). Scale bar: 100 μm. Source data are provided as a source data file.

tube explants cultures from E9.5 embryos performed under conditions favoring melanocytes proliferation and differentiation revealed a drastic reduction in the number of melanin-producing melanocytes in mutants compared to controls (Fig. 7c, ctl line, mean ± SEM of $167 \pm 42$ and $13 \pm 12$ melanocytes counted in six controls and seven mutants respectively,***$p < 0.001$, $t$ test). As for SCs, *Ifih1*-ShRNAs infection increased the number of melanin-producing cells in *Adar1* mutants ($49 \pm 12$, **$p < 0.01$; Fig. 7c), suggesting that concomitant deletion of Mda5 at least partially rescue pigmentation defects. In light of these results, the phenotype of *Adar1;Mavs*[11] and *Adar1;Ifih1* double mutants (ref. [45] and D. Stetson, personal communication) was re-interpreted and revealed no pigmentation anomalies in either models, suggesting that pigmentation defects we observed in vitro and in vivo result from a mechanism that is MDA5/MAVS dependent.

## Discussion

We used a conditional deletion strategy in mice to study the role of *Adar1* and A-to-I RNA editing during NC development. The first alteration we observed in mutants was global depigmentation. This was true irrespective of the NC-specific Cre driver line used (*HtPA-Cre* or *Wnt1-Cre*). RT-qPCR experiments using various melanocytic markers and use of the *R26R* allele under the control of the NC-specific Cre recombinase to trace YFP-positive melanocytes revealed that the early steps of melanoblast development are unaffected. However, there is a severe reduction in the number of melanocytes within the hair follicles of mutants from E18.5, resulting from the rapid death of these cells, and following upregulation of an ISGs signature recently shown to be activated upon *Adar1* deletion/RNA editing activity alteration in other tissues. As when *Adar1* is deleted in hematopoietic and

neuronal progenitors, massive cell death of melanocytes therefore explains the depigmentation observed[1,5,7].

NC *Adar1* deletion also severely alters SCs differentiation, i.e. no myelin is formed. Again, this observation was made using both NC-specific Cre drivers. Electron microscopy, RT-qPCR, and RNA-seq analyses showed that the mutant SCs are blocked at the promyelinating stage. Limited impact on SCs number and no difference in axonal diameter were observed, suggesting no evident alterations of neurons. This phenotype recalls that of *Egr2*, *Dicer*, and *Dgcr8* mutants[42,46–48], in which most of the SCs stall at the promyelinating stage, are unable to myelinate, and some radial sorting defects are observed. In *Dicer* and *Dgcr8* mutants, artificially sustained expression of genes characteristic of immature/promyelinating cells is proposed to explain the defects. Both mutants indeed display decreased *Egr2* expression and increased expression of *Sox2*, *Jun* and *Notch*, *Hes1*, *Egr1*, *Egr3*,*Ngfr*, *Ncam1*, and *Ccnd1*[42,46–48]. In our case, the expression of these genes was not significantly altered, except for *Egr1* and *Hes1*, which were upregulated in the *Adar1* mutants relative to controls. Although the phenotypes are very similar, these observations suggest that the underlying molecular mechanism is different. In the mutants presented here, additional deregulated expression of *Pou3f1*, *Pou3f2*, *Tfap2a*, *Tfap2b*, *Myc*, and *Cebps* transcription regulators known to play key roles in SCs or early NC development was noticed. Whether their deregulation is a cause or a consequence of the observed alterations is under investigation.

Interestingly, 34% of the RNA transcripts deregulated upon *Adar1* deletion are also deregulated upon nerve injury[39,40]. Injury-induced SCs reprogramming involves the downregulation of myelin genes and upregulation of immature SCs markers. In our case, however, some (but not all) early markers were more likely maintained and the myelin genes not induced. An additional specific molecular signature appears during injury, including (1) upregulation of neurotrophic factors promoting survival of the injured neurons; (2) upregulation of the expression of cytokines as part of the innate immune response; and (3) activation of a cell-intrinsic myelin breakdown process (autophagy)[25]. We also observed increased expression of *Shh*, *Gdnf*, *Nrg1 Artemin*, *Fgf5*, *Vgf*, *Btc*,*Tnf*, and *Edn1* in *Adar1* mutants. Because some of these (e.g. *Nrg1* and *Edn1*) are known to support SCs survival[24,49,50], this upregulation may explain why *Adar1* mutant SCs escape the massive cell death encountered in melanocytes. In our case, the inflammatory response was triggered by A-to-I RNA editing alterations (see below), but no or limited recruitment of immune cells was observed. Of note, *Dgcr8* mutants, but not *Egr2* mutants, also show upregulation of an injury-related gene expression program[42]. In addition to epigenetic mechanisms[28,38,39], RNA editing could therefore be involved in the control of injury-related gene expression programs. As for the *Dicer* and *Dgcr8* mutants, it would be of specific interest to test the function of ADAR1 in myelin maintenance under standard conditions or stress paradigms.

As in other tissues, analysis of the differentially expressed mRNAs using the interferome database additionally showed 75% of them to be within or associated with IFN type 1 or 2 signaling pathways[34]. However, analysis of liver from mutant mice revealed ISGs signature is not systemic, suggesting that activation of at least part of these ISGs is intrinsic to/specific for NC derivatives. SCs and melanocytes, which are typically nonimmune cells, indeed possess the capacity to elicit innate immune response by expressing a large number of innate immune genes in response to infections[51,52]. It is however difficult to ascertain the causality of such a large number of genes and identify primary versus secondary events in the phenotype genesis. Nevertheless, some of these ISGs are early NC expressed genes or negative regulators of myelination, and/or are part of the injured response described

above, suggesting that chronic activation of some of them may dysregulate the balance between positive and negative regulators of myelination, explaining the observed blockade of myelination. Of interest, 61 genes known to be transiently expressed after injury are abnormally maintained in mutants with a fold-change > 5, including *Soat2*, *Vgf*, *Egr1*, *Ch25h*, which have known functions in myelination/demyelination processes (Supplementary Data 2, genes in red). Their deregulation could therefore participate to the phenotype genesis.

Sixty A-to-I (G) editing sites were also specifically altered in mutants within or near 57 different genes. Whereas genome-wide analysis of the in vivo substrates of ADAR1 previously led to the identification of clustered editing within long dsRNA stem loops within the 3′ untranslated regions of endogenous transcripts[9,10], no such enrichment was observed here. Association between each of the concerned genes and SCs alterations seems also very limited. Although the identification of unedited RNA at the origin of any of the observed phenotypes will require further validation, we propose that their recognition by MDA5 is (as in other tissues) part of the pathogenic pathway[10]. Indeed, (i) extinction of *Ifih1* partially rescued the differentiation of *Adar1* mutant SCs in vitro, (ii) Sh-*Ifih1* at least partially rescued melanocytes defects of *Adar1* mutants in vitro, (iii) reinterpretation of the phenotype of *Adar1;Mavs* and *Adar1;Ifih1* double mutants in light of our results revealed no pigmentation anomalies (refs. [11,12,45] and D. Stetson, personal communication).

Although both Cre lines used allow deletion of *Adar1* in a large number of NC derivatives, it is striking that the alterations mainly affect melanocytes and SCs (no major or incomplete penetrant alterations of other NC structures identified to date) and could be observed only from E18.5 onwards. The trans-regulators that trigger the deregulation of RNA editing or the activation of cell-specific RNA species that initiate MDA5-mediated IFN production from E18.5 remain to be identified.

Beyond the description of the mouse phenotype, our results could help further extend the clinical spectrum associated to *ADAR1* mutations in human. Only one patient with AGS due to *ADAR1* mutation has been reported to present with associated peripheral neuropathy[13]. Data presented here suggest that neuropathy could have been overlooked considering the extreme disability of most patients. An oriented neurological evaluation of patients with AGS syndrome and *ADAR1* mutations is therefore needed to answer this question.

Our study also further expands our knowledge of the processes that drive DSH in humans[16,19,53,54]. Although the number of DSH patients with *ADAR1* mutations is increasing, very few studies have focused on the origin of the observed alterations[19,53,54]. Two reports showed a decreased number of melanocytes in hypo-pigmented regions[19,55] along with immature melanosomes, numerous vacuoles and signs of degenerative mitochondria[53]. We show here that the depigmentation identified upon *Adar1* deletion is due to a massive reduction in the number of melanocytes through apoptosis. Our results also recall observations made in the hyper-pigmented regions of DSH patients. In these regions, the melanocytes were reported to be enlarged, containing numerous and elongated dendrites, said to increase their functionality. In the present mouse model, the very few surviving *Adar1* mutant melanocytes were also enlarged relative to controls and produce melanin (legend of Fig. 2d). Whether their functionality is modified is yet to be determined. Despite these overall similarities, timing of the pigmentation defects seems to differ between species (depigmentation during infancy in humans versus loss of melanocytes before birth in mice). However, recent publications suggest that *ADAR1* mutations can also be associated with DSH from birth[56]. Aside DSH, similar findings are frequently observed in other inflammation-associated

pigmentary disorders, including vitiligo[57], opening the possible involvement of *ADAR1* and alterations of A-to-I editing in this disorder.

In conclusion, we highlight the function of ADAR1 in the development of melanocytes and SCs. Overall, our results suggest that this enzyme safeguards these NC-derived cells from unwanted MDA5-mediated IFN production and chronic ISG upregulation, with implications for the human diseases caused by *ADAR1* mutations, NC development and NC-linked-disorders.

## Methods

**Mice, genotyping and global phenotype analysis.** The mouse models used in this study were: (i) B6.129-Adar^tm1knk^/Mmjax, a model purchased from the Jackson Laboratory (stock number 34619-JAX, here referred as Adar1^fl/fl^), (ii) Gt (ROSA)26Sortm1(EYFP)Cos[32] (referred as R26R); (iii) Tg(PLAT-Cre)116Sdu[31] (referred as HtPA-Cre), and (iv) Wnt1-Cre driver[33].

Cre-floxed systems for gene deletion were used to get animals of interest. Breeding of *Adar1^fl/fl^* with R26R was first performed to obtain animals homozygous at both loci. Those were subsequently crossed with HtPA-Cre or Wnt1-Cre drivers to generate *HtPA-Cre;Adar1^fl/+^;R26R* or *Wnt1-Cre;Adar1^fl/+^; R26R* animals. Each of these were then crossed with *Adar1^fl/fl^;R26R* to generate progenies of different genotypes (*HtPA-Cre; Adar1^fl/fl^;R26R* or *Wnt1-Cre; Adar1^fl/fl^; R26R*) referred as mutants; wild-type (*Adar1^fl/fl^;R26R* or *Adar1^fl/+^;R26R*) and heterozygous for *Adar1* (*HtPA-Cre; Adar1^fl/+^;R26R* or *Wnt1-Cre;Adar1^fl/+^R26R*). Tails biopsies were used to perform DNA extraction making use of the direct PCR lysis reagent for Tail (Viagen) and subsequent genotyping. All primers purchased from Eurogentec and PCR primers are reported in Supplementary Table 2. The study complies with all relevant ethical regulations for animal testing and research. Indeed, experiments were done in accordance with the Institutional Animal and Use Committee guidelines of the Institut National de la Santé et de la Recherche Médicale (INSERM). The study received ethical approval from the Comite d'Ethique pour l'experimentation Animale (C2EA-12-035 and 16-097) and deposited under APAFIS 9783 number 2017022215395397.

**Histology and electronic microscopy.** Tissues (sciatic nerves, skin, gut and heart) were dissected and fixed in 4% paraformaldehyde, paraffin-embedded cuts and stained with hematoxylin and eosin. Alternatively, fixed tissues were frozen in Optimal Cutting Temperature media (OCT, Sakura), cut into 8 μm sections and used for immunofluorescence experiments. Skeletal preparations were performed on P4 controls and mutant embryos by Alcian Blue and Alizarin red. Briefly, the mice at P4 were eviscerated, fixed overnight in ethanol and then macerated in acetone for 24 h. The fixed fetuses were stained with Alcian Blue 8GX (150 mg/l in 80% ethanol, 20% acetic acid) for 48 h at 37 °C, washed in 70% ethanol for 8 h and cleared in 1% KOH overnight. They were then incubated in Alizarin Red S (50 mg/l in 2% KOH) for 6–8 h, cleared in 1% KOH and stored in 50% glycerol before observation.

For electron microscopy, mutant and control sciatic nerve samples were collected and fixed at 4 °C in 2.5% glutaraldehyde in sodium cacodylate buffer, rinsed and post fixed at 4 °C in a 1% osmium tetroxide solution. Samples were then dehydrated in graded alcohol and acetone baths and embedded in epoxy resin. After polymerization (for 48 h at 60 °C), samples were removed. Semi-thin transverse sections (1.5 μm) were stained with toluidine blue and ultrathin sections were stained with uranyl acetate and lead citrate and observed through a JEOL electron microscope. The number of myelinated nerve fibers in each nerves was counted manually on semi-thin transverse sections. The SCs count was performed manually based on photographs of nerve biopsy (electron microscopy) of control and mutant mice at each stage of development and reported to the surface studied to determine a density of cells per mm². On these same EM pictures, after having identified myelinated nerve fibers (and nerve fibers in the process of myelinating), the size of their axons was measured for each of them using ImageJ freeware (National Institutes of Health, USA).

**Immunofluorescence and immunohistochemistry.** Frozen sections were blocked in PBS + 0.1% Triton X-100 (PBT) for 1 h. Primary antibodies were diluted in blocking solution (PBT + 1% BSA + 0.15% glycine) as follows: cleaved-Caspase-3 (casp3, D175, lot 43, rabbit, Cell Signaling Technologies; 1:200 dilution), Mbp (SMI 94R-0100 lot E10172EF, mouse, Covance, 1:1000 dilution), GFP (A11122, lot 1891900, rabbit, Invitrogen, 1:1000 dilution), CD45 (14-045-185 C30F11, Rat, Invitrogen, 1:100 dilution) and F4/80, CD11b, CD68 cocktail (MCA497, MCA74G, MCA1957, Rat, Bio-Rad, 1:200 dilution) and incubation performed overnight at 4 °C. After several washes, slides were incubated with secondary antibody (anti-rabbit alexaFluor555, anti-mouse alexaFluor488, anti-rat Cy3 or anti-rabbit alexaFluor488, Invitrogen, 1:500 dilution each) for 2 h before final washing. For dorsal root ganglia (DRGs) cultures, cells were fixed in 4% PFA for 10 min and immunostained using the same procedure with TuJ1 (MMS-435P, lot TU17044, mouse, Eurogentec, 1:1000 dilution) and Mbp (ab40390, Rabbit, Abcam, 1:200 dilution). Secondary antibodies used were as follows: anti-mouse alexaFluor 488 and anti-rabbit alexaFluor 555

(Invitrogen, 1:500 dilution each). Preparations were then mounted using Vectashield containing DAPI (Vector laboratories) and observed using Spinning Disk from Zeiss.

**Quantitative real-time PCR.** Skin, sciatic nerves, heart, gut, mandible and liver from P4 and newborn animals or E16.5 and E18.5 embryos of different genotypes were dissected and total RNA extracted using RNeasy Mini or Micro kit (Qiagen) according to the manufacturer's instructions. After DNAse treatment, concentration and quality of RNA was determined using the NanoDrop (Thermofisher) and Xpose (Trinean) apparatus. cDNA was synthesized using the Maxima First Strand cDNA Synthesis. Quantitative real-time PCR (qPCR) was performed using the Maxima SYBER green/ROX qPCR Master Mix X2 (all from Thermofisher) and amplified using Mastercycler® RealPlex² (Eppendorf). The relative abundance values of each amplification product were normalized to the internal control β-actin, and mRNA expression levels in mutants expressed relative to controls. PCR primers are reported in Supplementary Table 3.

**RNA-seq library preparation and sequencing.** Total RNAs of sciatic nerves were isolated using the RNeasy Kit (Qiagen) including a DNAse treatment step. RNA quality was assessed using RNA Screen Tape 6000 Pico LabChips with the Tape Station (Agilent Technologies) and the RNA concentration was measured by spectrophotometry using Xpose. RNA-seq libraries were prepared starting from 60 ng of total RNA using the Universal Plus mRNA-Seq (Nugen) as recommended by the manufacturer. The oriented cDNA produced from the poly-A + fraction were PCR amplified (15 cycles). An equimolar pool of the final indexed RNA-Seq libraries was sequenced on an Illumina HiSeq2500 (paired-end reads 130 bases + 130 bases). A mean of 124 millions of passing filter paired-end reads was produced per library sample (with a minimum of 103 million and a maximum 158 millions).

**RNA-seq alignment and bioinformatics data analysis.** RNA-Seq data analyses were performed by GenoSplice [https://www.genosplice.com/]. Sequencing, data quality, reads repartition (e.g., for potential ribosomal contamination), and insert size estimation were performed using FastQC, Picard-Tools, Samtools and rseqc. Reads were mapped using STARv2.4.0[58] on the mm10 Mouse genome assembly. Gene expression regulation study was performed following classic scheme[59–61]. Briefly, for each gene present in the Mouse FAST DB v2018_1 annotations, reads aligning on constitutive regions (that were not prone to alternative splicing) were counted. Based on these read counts, normalization and differential gene expression were performed using DESeq2[62] on R (v.3.2.5). Only genes expressed in at least one of the compared experimental conditions were further analyzed. Genes were considered as expressed if their rpkm value was greater than 97.5% the background rpkm value based on intergenic regions. Results were considered statistically significant for *p* values ≤ 0.05 and fold-changes ≥ 2.0 in mutants compared to controls. 20.1% of the expressed genes (3009/14402) were thus found up- or downregulated in mutants versus controls. GO term and pathway analysis were subsequently performed on this list, making use of KEGG, REACTOME, and Ingenuity pathway analysis. We sought A-to-I (G) RNA variants using union of results from GATK HaplotypeCaller and RED-ML[63] on minimal total coverage in all samples ≥ 10 reads. A to G RNA variants were annotated using VEP and filtered if they correspond to already known genomic variant from dbSNP142, and retained if % editing level was significantly reduced in mutants compared to control samples (*p* ≤ 0.1). To check if editing sites were previously reported, we used the RADAR or DARNED databases [https://rnaedit.com] and [https://darned.ucc.ie]. Several of these were found similarly altered in independent set of four control and mutant samples.

**Mixed SCs and NC cultures and rescue experiments.** DRGs cultures were prepared from isolated DRGs of E13.5 embryos. Briefly, dissociated cells were plated at a density of $4 \times 10^4$ cells per 13 mm coverslip coated with Matrigel (BD Biosciences) and poly-D-Lysine (Sigma). Cells were kept in neurobasal medium supplemented with B-27, penicillin/streptomycin, glutamax (from Gibco life Technologies) and 50 ng/ml NGF (PreproTech) for 2 days, and then in basal medium supplemented with gluta-max, D-glucose, BSA, P/S, ITS and NGF for seven additional days as in ref. [44]. Cells were then maintained for 10–13 additional days in medium supplemented with 50 μg/ml L-ascorbic acid (Sigma), 15% heat inactivated calf serum and 5 μg/ml heparin (Sigma) to induce myelination, before fixation.

Neural tube explants were obtained from E9.5 embryos and plated on six-well plates coated with 20 μg/ml of human plasma fibronectin (GIBCO BRL) diluted in PBS. The basal culture medium consisted of F12 medium supplemented with 10% fetal calf serum, 16 nM TPA (Sigma), 2 μg/ml [Nle4, D-Phe7]-a-MSH (Sigma) and Edn3 at a concentration of 3 nM[64]. Medium was changed every second to third day for a period of 21 days to allow melanocytes differentiation and melanin production before fixation.

Three days after neural tube explants seeding or before myelination induction, cells were infected with pLKO ShRNA targeting *Ifih1* (TRCN0000103646: 5′-CCACAGAATCAGACACAAGTT-3′) or control ShRNA (SHC002V) (Sigma) with a multiplicity of infection (MOI) of 2. Infection was performed using standard procedure. We chose this strategy because lentiviral ShRNA-mediated KO is efficient in SCs and this specific Sh-*Ifih1* was previously used with success in *Adar1* mutant hematopoietic stem and progenitor cells[10].

**Statistical analysis**. All analyses were done using Graphpad Prism 6.0. Data are presented as mean ± SEM in the graphs and were analyzed using unpaired two-tailed Student's $t$ test for simple comparison or one-way ANOVA for multiple comparisons. $p < 0.05$ was considered statistically significant. $p$ values were as follows: not significant (ns) >0.05, *<0.05, **<0.01, ***<0.001. Fisher Exact test was used to delineate data overlap presented in Venn diagrams of Figs. 5 and 6. For distribution plots presented in Fig. 4 and Supplementary Fig. 4, the Kolmogorov–Smirnov test was used for statistical analysis.

**Reporting summary**. Further information on research design is available in the Nature Research Reporting Summary linked to this article.

## Data availability

The data that support the findings of this study are all available within the text, supplementary, or source data files. The source data underlying Figs. 2b, c, 3a–e, 4d–f, 5b, 6b, 7b, c, and Supplementary Figs. 1a, 2, 3, 4c, 5a–c, 6b, 7a–c, 8 are provided as a Source Data file. The RNA-seq datasets generated during the study have been submitted to the [GEO] repository and accepted under [GSE127795]. The datasets reanalyzed for comparison purpose are [E-MTAB-5633] in Array Express [https://www.ebi.ac.uk/gxa/experiments/EMTAB-5633?ref=aebrowse] and [GEO] database [accession #GSE33454] provided in refs. [39,40].

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

## Acknowledgements

The authors would like to thank members of the IMRB and LEAT-Imagine platforms for animal husbandry, members of the histology platform of Broussais Hospital for technical help, Yuli Watanabe and Edouard Reyez Gomez for help at the beginning of the project, Benoit Funalot and Yanick Crow for advices, Olivier Ariste and Ariane Jolly for help in data analysis, Elisabeth Dupin Veronique Delmas and Lionel Larue for advices on melanocytes rescue experiments, Frederic Rieux-Laucat for advices on immune cells recruitment and Brahim Nait Oumesmar for CD45 and (F4/80+CD11b+CD68) antibodies. This work was supported by the Fondation ARC pour la recherche sur le cancer and AFM trampoline grant to N.B., Institut National Pour la Santé de la Recherche Médicale (INSERM), state funding from the Agence Nationale de la Recherche under the "Investissements d'avenir" program [ANR-10-IAHU-01] and the MSDAvenir fund [DEVO—DECODE project]. N.G. and A.K. both received a 3-year Ph.D. fellowship from ED-SVS402, L.Z. received a 3-year Ph.D. fellowship from ED-BioSPC. N.G. is a beneficiary of an Institut Imagine 4th-year Ph.D. fellowship (7 months) from Fondation Bettencourt Schueller and supported by Fondation des treilles.

## Author contributions

N.G., A.K. and N.B. conceived, designed and performed the experiments and analyzed data; N.G. and N.B. wrote the manuscript and prepared figures; N.B. supervised the study and obtained ARC and AFM funding; L.Z. performed validation of RNA-seq data by RT-qPCR and helped with rescue experiments as well as immune cells and systemic response experiments; L.R. prepared samples for electronic microscopy; S.M. performed axon diameter quantification; R.P.K. analyzed gut histological sections; M.P. prepared libraries and performed sequencing; P.D.L.G. performed bioinformatics analysis of RNA-seq data; J.M.V. prepared and analyzed electronic microscopy data; S.D. provided the HtPA-Cre line; V.P. discussed results; V.P., S.D., L.Z., R.P.K., J.A. made critical revision of the manuscript and all authors approved the manuscript.

## Competing interests

The authors declare no competing interests.
