## [Peer Review File · Nature Communications]

Reviewers' comments:

Reviewer #1 (Remarks to the Author):

The manuscript by Gacem et al., entitled "ADAR1-mediated regulation of neural crest-derived melanocytes and Schwann-cell development", describes the role of the RNA editor Adar1 (adenosine deaminase acting on double stranded RNA 1) in the differentiation of two neural crest derivatives, melanocytes and myelinating Schwann cells in mouse embryonic development.

Gacem et al show that the RNA editor ADAR1 is required for melanocyte survival and Schwann cell myelination in early development and examine the protective role of the RNA sensor MDA5 in interferon-mediated innate immune response upon ADAR1 loss of function. This work is consistent with previous studies demonstrating the implication of MDA5 in autoimmune and autoinflammatory diseases such as Aicardi-Goutières syndrome and better elucidates the molecular mechanisms responsible for defects such as depigmentation and motor defects in human disorders caused by mutations of ADAR1. This article depicts a very interesting finding where the detection of unedited RNA triggers an innate immune response transcriptomically similar to nerve injury that precedes and prevents Schwann cell differentiation into myelinating Schwann cells.

I will first make general comments on the manuscript. None of the immunohistochemistry figures have scale bars or are quantified, the latter is a big disadvantage for the study. It is a bit disappointing that the literature is not always thoroughly cited and some paragraphs are only superficially supported by references to previous studies.

Although the authors have undertaken a very helpful electron microscopy ultrastructural analysis of peripheral myelin, they haven't fully explored their dataset, and one caveat I see is the lack of quantification of axonal diameter. The fact that the homozygous pups present a much lower body weight at post-natal day 4 (figure 1c) leads me to hypothesize that they may also have smaller axons or likely more small caliber axons than their WT counterparts, which can be difficult to appreciate in the raw electron micrographs (showed in figure 4a and supplemental figures). Along this line, it is not clear whether the mutant pups present a developmental delay and catch up in weight with their WT siblings at a later stage (e.g. P8 or P10), or maintain that weight difference throughout the ages examined in the study. For this reason, a further analysis of the EM data would greatly help support the conclusions the authors made from their in vivo analysis.

Another example of regrettable absence of quantification and statistical analysis is shown in figures 3b and 4b. The authors used an activated-caspase 3 antibody to assess melanocyte apoptosis in skin sections of E18.5 mutants and Schwann cell apoptosis in sciatic nerves of P4 animals, respectively. I think quantifications of the loss of melanocytes are needed to illustrate the immunostaining and should be included in the manuscript.

Regarding the Schwann cell apoptosis assay presented in figure 4b, my concern is that the magnification might not be high enough to allow for evaluation and/or quantification of apoptotic cells. Along the same line, although it is not statistically significant, both Sox10 and Egr2 mRNA levels are decreased by ~25% in P4 mutant sciatic nerves (figure 4c) and there are generally less Schwann cell nuclei detectable in the electron micrograph (figure 4a ii-2). This makes me wonder whether there could be less Schwann cells in the Adar1 mutants.

The transcriptomic analysis revealing the nerve injury signature is very nice and its validation by RT-qPCR is a considerable work that I believe is the most robust and valuable part of the article.

I think it is important to also highlight the effort that the authors put into this study by examining two convincing conditional knock-out models of Adar1 loss of function in the neural crest lineage using Wnt1 and human tissue plasminogen Cre.

Last, whereas the authors have successfully rescued Schwann cell differentiation in vitro by invalidating MDA5 (figure 7), the melanocyte phenotype remains left hanging and gives an impression of unfinished work.

Reviewer #2 (Remarks to the Author):

Key results:

The neural crest KO of Adar1 generates a phenotype related to a failure in melanocyte and schwann cell survival/function.

Validity:

the work is well performed and controlled. Several Cre's and approaches are utilised and orthogonal approaches. The work is highly consistent and supportive of a significant number of previous publications related to the function of Adar1.

Originality and significance:

Whilst a well performed and controlled study, the results are largely what could have been anticipated based on a number of other papers assessing Adar1 function in mouse models.

Data & methodology: The results and interpretation reflect the findings. Appropriate cohort sizes and methods appear to have been applied to support the conclusions.

Suggested improvements:

The findings are consistent with what would have been anticipated. It is difficult to interpret the findings on differential gene expression given the profound impact of the innate immune response on the gene expression patterns and subsequent cell development. This could potentially be commented on.

The text uses "invalidation" to describe the deletion of the Adar1 gene - I find this term an interesting choice and to me slightly distracting.

The results are well presented and clearly described. The discussion could be shortened without negatively impacting the work.

Reference to Mannion et al Cell Reports 2014 Nov 20;9(4):1482-94 should be added to ref 10 and 11.

Clarity and context:

this is appropriate.

Reviewer #3 (Remarks to the Author):

In this paper, Gacem and colleagues address the role of Adar1 in neural crest development by Cre-dependent conditional gene ablation. Adar1 mediates A to I RNA editing associated with various forms of posttranscriptional regulation. Other studies have previously shown that Adar1 inactivation results in activation of an interferon response independently of bacterial or viral infections. Moreover, ADAR1 mutations have been linked to human disorders involving, among others, pigmentation defects. Thus, investigating Adar1 functions in neural crest/melanocyte development is of interest as such.

The present study reveals that conditional Adar1 deletion in the neural crest affects Schwann cell differentiation and melanocyte survival. However, it leaves open many questions associated with neural crest-specific Adar1 deletion and does not provide any molecular mechanisms underlying the observed phenotypes. In particular, it remains to be addressed

whether neural crest derivatives other than Schwann cells and melanocytes are affected by Adar1 inactivation, i.e. whether there is a general defect in neural crest development;
why distinct neural crest derivatives are differentially affected by Adar1 inactivation (survival vs differentiation);
whether the observed phenotypes have anything to do with induced expression of interferon-stimulated genes and, importantly, with the innate immune system;
and whether Adar1-mediated A to I RNA editing is functionally implicated at all in neural crest development.

In sum, the present study is too preliminary for publication in this journal.

Reviewer #1: *The manuscript by Gacem et al., entitled “ADAR1-mediated regulation of neural crest-derived melanocytes and Schwann-cell development”, describes the role of the RNA editor Adar1 (adenosine deaminase acting on double stranded RNA 1) in the differentiation of two neural crest derivatives, melanocytes and myelinating Schwann cells in mouse embryonic development. Gacem et al show that the RNA editor ADAR1 is required for melanocyte survival and Schwann cell myelination in early development and examine the protective role of the RNA sensor MDA5 in interferon-mediated innate immune response upon ADAR1 loss of function. This work is consistent with previous studies demonstrating the implication of MDA5 in autoimmune and autoinflammatory diseases such as Aicardi–Goutières syndrome and better elucidates the molecular mechanisms responsible for defects such as depigmentation and motor defects in human disorders caused by mutations of ADAR1. This article depicts a very interesting finding where the detection of unedited RNA triggers an innate immune response transcriptomically similar to nerve injury that precedes and prevents Schwann cell differentiation into myelinating Schwann cells.*

I will first make general comments on the manuscript. None of the immunohistochemistry figures have scale bars or are quantified, the latter is a big disadvantage for the study. It is a bit disappointing that the literature is not always thoroughly cited and some paragraphs are only superficially supported by references to previous studies.

We thank the reviewer for pointing these important missing informations. We added scale bars on each immunohistochemistry figures and indicated corresponding scales in associated legends. We also performed quantifications required (see below) and added references to support findings presented in some sections (indicated in red in the references section). We limited references to 71 to follow recommendations of the journal. We apology in advance for still missing ones and are happy to add specific ones upon request.

Although the authors have undertaken a very helpful electron microscopy ultrastructural analysis of peripheral myelin, they haven't fully explored their dataset, and one caveat I see is the lack of quantification of axonal diameter. The fact that the homozygous pups present a much lower body weight at post-natal day 4 (figure 1c) leads me to hypothesize that they may also have smaller axons or likely more small caliber axons than their WT counterparts, which can be difficult to appreciate in the raw electron micrographs (showed in figure 4a and supplemental figures). Along this line, it is not clear whether the mutant pups present a developmental delay and catch up in weight with their WT siblings at a later stage (e.g. P8 or P10), or maintain that weight difference throughout the ages examined in the study. For this reason, a further analysis of the EM data would greatly help support the conclusions the authors made from their in vivo analysis.

In order to answer the comment raised by the reviewer, axon diameter quantification was performed on EM data from controls and mutant at post-natal days 4, 8 and 10. Results are presented in the text and the modified Figure 4 and Supplementary Figure 2. The raw data are available in the source data file. As requested in one of the other comments below, quantification of the number of Schwann cells was also performed. Axon diameter was similar in controls and mutants at all stages analyzed. Although a trend towards lower number of Schwann cells is observed in mutants relative to controls at all stages, significant statistical difference is only detected at P10.

Quantification of data at post-natal day 4 are presented in modified Figure 4b and in the results section. You can now read “In line with these observations, quantification revealed a drastic decrease in the number of myelinated axon profiles per nerve section in mutants compared to controls, while the number of Schwann cells per square millimeter and axon diameters were not significantly different in mutants compared to controls (Fig. 4b).”

Quantification of data at post-natal day 8 and 10 are presented in the results section and modified Supplementary Figure 2. You can now read “Quantification again confirmed the drastic decrease in the number of myelinated axon profiles per nerve section (this number reached up to 3,000 in controls, but never over 200 in the mutants). A significant (although modest) reduction in the number of Schwann cells per square millimeter was observed at P10 only. No change in axon diameter was observed at P8 and P10 (Supplementary Fig.2c). Altogether, results thus suggest a blockade of differentiation rather than a simple delay.”

We also clarified the description of mutant’ phenotype. Indeed, none of the mutants generated “catch up” later. In the result section you can now read “Four days later (post-natal day 4, P4), the heterozygous animals were still indiscernible from the wild-types (as in every stage analyzed). In contrast, the surviving mutants showed a statistically significant reduction in body weight and almost total lack of pigmentation (Fig.1c). From this stage onwards, mutants additionally developed tremors and an unsteady gate, were largely unresponsive to stimuli and runted. As shown upon survival rate analysis, all mutants died within the first 10 days of life (Fig.1d).”

Another example of regrettable absence of quantification and statistical analysis is shown in figures 3b and 4b. The authors used an activated-caspase 3 antibody to assess melanocyte apoptosis in skin sections of E18.5 mutants and Schwann cell apoptosis in sciatic nerves of P4 animals, respectively. I think quantifications of the loss of melanocytes are needed to illustrate the immunostaining and should be included in the manuscript.

Quantification of cell apoptosis of melanocytes and of Schwann cells have been performed. Raw data are presented in the source data file, mean values in controls and mutants along with statistical analysis presented in text and in the modified figures 3 and 4.

For melanocytes you can now read “Quantification revealed an average percentage of apoptotic YFP positive cells of 21.8 ± 0.9 % versus 71.4 ± 8.4 % in controls and mutants, respectively (Fig.3b), suggesting that the global depigmentation observed is due to the rapid loss of melanocytes by cell death.”

For Schwann cells you can now read “We observed only few apoptotic cells (activated Caspase3-positive cells) in the sciatic nerves of mutants relative to controls. Quantification confirmed that although significantly increased in mutants relative to controls, percentage of apoptotic cells did not exceed 8% (1.34 ± 0.27 % versus 7.71 ± 1.69 % apoptotic cells in controls and mutants, respectively, Fig.4c).”

Along the same line, although it is not statistically significant, both Sox10 and Egr2 mRNA levels are decreased by ~25% in P4 mutant sciatic nerves (figure 4c) and there are generally less Schwann cell nuclei detectable in the electron micrograph (figure 4a ii-2). This makes me wonder whether there could be less Schwann cells in the Adar1 mutants.

Quantification of Schwann cell number was performed at postnatal day 4, 8 and 10. Results are presented in Figure 4b and supplementary Figure 2 and discussed along with axon diameter (see answer to previous comment).

As mentioned by the reviewer a trend towards lower number of Schwann cells is indeed observed at all stages in mutants relative to controls. These quantifications are in perfect correlation with the 25% decrease in *Sox10* and *Egr2* expression levels and with the limited (7%) increase in cell death observed at P4. However, because significant statistical difference is only detected at P10, these differences are discussed with precaution.

Within the discussion section you can now read “Limited impact on Schwann cell number and no differences in axonal diameter were observed, suggesting no evident alterations of neurons (small or large caliber).”

The transcriptomic analysis revealing the nerve injury signature is very nice and its validation by RT-qPCR is a considerable work that I believe is the most robust and valuable part of the article. I think it is important to also highlight the effort that the authors put into this study by examining two convincing conditional knock-out models of Adar1 loss of function in the neural crest lineage using Wnt1 and human tissue plasminogen Cre.

We thank the reviewer for this comment and would like to mention that raw data corresponding to RTqPCR validation are now presented in in the source data file.

Last, whereas the authors have successfully rescued Schwann cell differentiation in vitro by invalidating MDA5 (figure 7), the melanocyte phenotype remains left hanging and gives an impression of unfinished work.

In the first version of the manuscript, we did not perform MDA5 rescue experiments in melanocytes because, although not interpreted as such by the authors, reevaluation of published data suggested that depigmentation indeed result from a mechanism that is MDA5/MAVS dependent.

As stated in the former first paragraph of our discussion: “...Pigmentation anomalies could not be evaluated in conventional *Adar1* single mutants due to the early embryonic lethality, but our study clearly shows that the survival of melanocytes is markedly altered upon *Adar1* invalidation in NC cells. Concomitant invalidation of *Ifih1* or *Mavs* rescued the embryonic lethality in the conventional model. Remarkably, reevaluation of the phenotype of these double mutants in light of our results revealed no pigmentation anomalies, suggesting the total rescue of melanocyte survival.”

To clarify and strengthen this point we contacted Dan Stetson (corresponding author of the manuscript entitled “Isoforms of RNA-Editing Enzyme ADAR1 Independently Control Nucleic Acid Sensor MDA5-Driven Autoimmunity and Multi-organ Development”) to discuss this specific point. He confirmed that, as shown in the mentioned paper, *Adar1;Mavs* double mutants do not present pigmentation alterations (Supplementary video available at

<https://www.sciencedirect.com/science/article/pii/S1074761315004537?via%3Dihub#mmc3>). He also confirmed that they never observed differences in phenotype between *Adar;Mavs* double knockout mice and *Adar;Ifih1* double knockout mice. In particular, these double mutants have no pigmentation defects, despite their early postnatal lethality.

In parallel, we performed in vitro rescue experiments. To this end, neural tubes from the trunk region were dissected from E9/E9.5 wild type or mutant embryos, plated on fibronectin coated dishes in specific medium favoring migration of neural crest cells away from the neural tube and their subsequent differentiation into melanocytes. We chose this system because isolation of NC progenitors as they emigrate from the neural tube is known to provide optimal preservation of NC cell potential and migratory properties. As in the case of Schwann cells cultures, our results show that this system nicely reproduced the pigmentation defects observed in mutants: i.e. drastic reduction of melanin expressing cells is observed in mutant cultures compared to controls. Remarkably, *ShMda5* infection at least partially rescued defects observed.

These in vitro experiments are now presented at the end of the results section entitled “Identification of RNAs targeted by editing and rescue experiments: downregulation of *Mda5* rescues both Schwann cells and

melanocytes defects in *Adar1* mutants.” and accompanying figure 7 (modified to include melanocytes rescue experiments). Methods section was also modified to include this in vitro culture system as well as adequate reference.

Both Dan Stetson in vivo observations (published and personal communication) and our in vitro experiments are also discussed in a new paragraph regrouping Schwann cells and melanocytes rescue upon Mda5/Mavs invalidation (now page 21 of discussion) to attract attention on this particular important information.

Reviewer #2:

Key results: The neural crest KO of Adar1 generates a phenotype related to a failure in melanocyte and schwann cell survival/function.

Validity: the work is well performed and controlled. Several Cre's and approaches are utilised and orthogonal approaches. The work is highly consistent and supportive of a significant number of previous publications related to the function of Adar1

Originality and significance: Whilst a well performed and controlled study, the results are largely what could have been anticipated based on a number of other papers assessing Adar1 function in mouse models.

Data & methodology: The results and interpretation reflect the findings. Appropriate cohort sizes and methods appear to have been applied to support the conclusions.

Clarity and context: this is appropriate.

Suggested improvements:

The findings are consistent with what would have been anticipated.

This work is indeed consistent with previous studies demonstrating that ADAR1, through its editing activity, safeguards cells from aberrant MDA5-mediated interferon production. Although the mechanism driven by *Adar1* invalidation is already known, our results extend the landscape of ADAR1 function to the field of neural crest development. It also demonstrates that in a manner similar to other cell types or tissues, *Adar1* deletion leads to massive cell death of melanocytes. However, the finding that a different mechanism is at work in Schwann cells is of great interest. We propose that the 'repair Schwann cell' state that we have identified alters their differentiation but maintains their viability. This specific signature raises the possibility of the involvement of ADAR1 and RNA editing in the control of repair processes, at least in the peripheral nervous system.

By showing specific alterations of melanocytes and Schwann cells, without drastic alterations of other neural crest derivatives, our results also help better elucidate the molecular mechanisms responsible for depigmentation and motor defects in human disorders caused by *ADAR1* mutations.

To strengthen these points, we reorganized the discussion to focus on 1) effect of *Adar1* invalidation on neural crest derivatives in mouse and their consequences, 2) impact of our results to the knowledge of the processes that drive *ADAR1* related human pathologies.

It is difficult to interpret the findings on differential gene expression given the profound impact of the innate immune response on the gene expression patterns and subsequent cell development. This could potentially be commented on.

Upregulation of ISG expression is indeed by far the highest enrichment observed in mutants compared to controls. Analysis of the 3,009 differentially expressed mRNAs showed 75% of them to be within or associated with IFN type 1 or 2 signaling pathways. The majority of these are probably secondary events with or without specific consequences in regards to the phenotype observed and indeed complicate the interpretation of data.

Nevertheless, as mentioned by reviewer 1, our RNA-seq data also revealed that detection of unedited RNA triggers an innate immune response transcriptomically similar to nerve injury that precedes and prevents Schwann cell differentiation into myelinating Schwann cells. These similarities led us to propose that aberrant expression/maintenance of some of these ISG (especially if they are also involved in early phases of repair process), may dysregulate the balance between positive and negative regulators of myelination, explaining the block of myelination observed. Comparison of list of genes we present in Supplementary Table 2 (common genes in repair versus our data, known to be ISGs) could therefore help identify some of the deregulated genes at the origin of phenotype observed.

To answer the reviewer concerns, we modified the following paragraph within discussion: “Analysis of the 3,009 differentially-expressed mRNAs using the interferome database showed 75% of them to be within or associated with IFN type 1 or 2 signaling pathways. It is difficult to ascertain the causality of such a large number of genes and identify primary versus secondary events. However, some of these are early NC-cell genes or negative regulators of myelination, and/or are part of the injured response described above, suggesting that persistent expression of some of these (due to their activation by the innate immune system response chronically activated in the mutants) may dysregulate the balance between positive and negative regulators of myelination, explaining the observed blockade of myelination. In line with this hypothesis, chronic upregulation or mutation of certain ISGs have already been shown to be involved in the progression of various disorders (upregulation of CH25H in Alzheimer’s disease and atherosclerosis and mutations of *MDA5* and *DDX58* (encoding RIG1) in AGS).”

The text uses "invalidation" to describe the deletion of the Adar1 gene - I find this term an interesting choice and to me slightly distracting.

We replaced the term invalidation by deletion within the whole manuscript

The results are well presented and clearly described. The discussion could be shortened without negatively impacting the work.

We reorganized and shortened the discussion to answer concerns of all three reviewers

Reference to Mannion et al Cell Reports 2014 Nov 20;9(4):1482-94 should be added to ref 10 and 11.

The reference was added

Reviewer #3:

In this paper, Gacem and colleagues address the role of Adar1 in neural crest development by Cre-dependent conditional gene ablation. Adar1 mediates A to I RNA editing associated with various forms of posttranscriptional regulation. Other studies have previously shown that Adar1 inactivation results in activation of an interferon response independently of bacterial or viral infections. Moreover, ADAR1 mutations have been linked to human disorders involving, among others, pigmentation defects. Thus, investigating Adar1 functions in neural crest/melanocyte development is of interest as such.

The present study reveals that conditional Adar1 deletion in the neural crest affects Schwann cell differentiation and melanocyte survival. However, it leaves open many questions associated with neural crest-specific Adar1 deletion and does not provide any molecular mechanisms underlying the observed phenotypes. In particular, it remains to be addressed whether neural crest derivatives other than Schwann cells and

melanocytes are affected by Adar1 inactivation, i.e. whether there is a general defect in neural crest development;

To answer reviewer comment, we analyzed three other NC derivatives: enteric nervous system, heart NC derived structures and NC craniofacial derivatives. The results are presented in the text and the new Supplementary Figure 5. Macroscopic and histopathological analysis of the gut and heart of homozygous mutant mice at P4 revealed no obvious alterations compared to controls. Indeed, although stomach of mutants are always smaller, enteric ganglia are present in apparent normal number and position along the whole length of the gut. Analysis of the heart revealed no outflow tract defects. In parallel, we performed skeletal analysis by Alcian blue followed by Alizarin red coloration. Although heads of mutants were significantly smaller (but correlate with global size reduction of mutant pups), no specific facial alterations were noticed. Only a fraction of mutants (4 out of 13 analyzed, 30%) were presenting with cleft palate, suggesting some craniofacial alterations might be an incompletely penetrant feature.

We further performed RT-qPCR experiments on RNA extracted from dissected gut, heart or mandible of controls and mutants to analyze the ISG signature and the expression of three chosen NC expressed markers. Although an increase in ISG expression was observed in mutants relative to controls in all three tissues, no alteration of NC expressed genes was detected. These data therefore suggest no or rather limited impact of *Adar1* deletion on development of these three NC derivatives. Although we cannot exclude functional or cell type-specific alterations, our data suggest that *Adar1* deletion does not affect all NC derivatives equally.

These additional data are now presented in a subsection of the results entitled “Deletion of *Adar1* in NC cells does not affect all their derivatives equally” and in the new Supplementary Figure 5 and accompanying legend. A sentence was also added within the introduction (page 5) to present NC origin of enteric nervous system, and contribution of NC cells to heart and craniofacial structures. Supplementary table 5 also includes additional RT-qPCR primers.

why distinct neural crest derivatives are differentially affected by Adar1 inactivation (survival vs differentiation);

The massive cell death of melanocytes recalls those of most cells types analyzed upon *Adar1* deletion, including hematopoietic and neuronal progenitors.

Schwann cells response is however different. As mentioned earlier, we suggest that RNA-seq data we generated gives arguments to explain this difference. Indeed, 34% of the RNA transcripts deregulated upon *Adar1* invalidation are also deregulated upon nerve injury. In the latter, the up-regulated transcripts includes those encoding neurotrophic factors (including *Gdnf*, *Shh*, and *Nrg1*) promoting survival of the injured neurons and axonal elongation. A large number of neurotrophic factors are also upregulated in the sciatic nerves of *Adar1* mutants compared to controls, including *Nrg1*, *Shh*, *Gdnf*, *artemin*, *Fgf5*, *Vgf*, *Btc*, *Tnf*, *Edn1*. Upregulation of these factors, some of which known to support Schwann cell survival, might explain why *Adar1* mutant Schwann cells escape the massive cell death encountered in melanocytes.

This point is now discussed page 19.

whether the observed phenotypes have anything to do with induced expression of interferon-stimulated genes and, importantly, with the innate immune system; and whether Adar1-mediated A to I RNA editing is functionally implicated at all in neural crest development.

Published data suggest that the primary function of ADAR1 is A-to-I editing, a failure of which results in inappropriate activation of the innate immune system by endogenous, unedited, dsRNA. Alteration of editing

and activation of the interferon response are therefore intimately linked. Work from the group of Stetson et al also uncovered a MDA5-MAVS-independent function for ADAR1 in the development of multiple organs, including kidney, small intestine and lymph node.

As far as NC derivatives are concerned, our *in vitro* rescue experiments as well as reinterpretation of *Adar1;Ifih1* double mutants pigmentation phenotype both provide strong arguments in favor of aberrant activation of Mda5/Mavs as a major cause of the Schwann cells and melanocytes defects observed (see rescue experiments and answer to reviewer 1). Because MDA5 is the key sensor involved in recognition of unedited RNAs and is triggering innate immune system activation in other tissues, it suggests that a similar mechanism is taking place here.

To answer reviewer comment, we modified introduction page 3 and 4 to clarify editing/ Mda5/ ISG activation association/ causality as well as Mda5/Mavs independent function. A list of unedited RNAs in mutants compared to controls is presented in supplementary table 4.

We also reorganized the discussion to regroup rescue experiments in one paragraph and thus clarify and strengthen the interpretation of these experiments that are key to answer this concern (see page 20-21)

Reviewers' comments:

Reviewer #1 (Remarks to the Author):

The revised manuscript from Gacem et al. on the role of Adar1 in melanocyte survival and Schwann cell myelination in mouse embryonic development has considerably improved. One of the main caveats was that the data, although potentially robust, were not fully explored and described. This concern has been addressed with the rigorous analysis of the existing data and the addition of convincing quantifications. The Schwann cell myelination phenotype is strengthened as the authors were able to rule out the possibility of an axonal defect or a decrease in the number of Schwann cells. The conclusions presented in the manuscript that were hypothetical are now suitably supported by the results and quantifications.

Finally, the authors have nicely completed the story by pushing the melanocyte analysis further. New *in vitro* experiments rescuing melanocyte function have been performed and were added to the manuscript along with an expanded discussion clarifying the pigmentation anomalies observed in the Adar1 mutants.

Reviewer #2 (Remarks to the Author):

The authors have addressed the comments raised during review. Given the ultimate conclusion that this is an editing dependent, Mda5 dependent phenotype the reference that best demonstrates this is PMID: 28874170 rather than relying on the unpublished perinatal data/observations from the Stetson lab.

Reviewer #3 (Remarks to the Author):

The authors performed new experiments and provide further data to strengthen their study. However, I still feel that more mechanistic insights into how Adar1 controls neural crest development would be required for publication in this journal.

As mentioned before, others have previously linked Adar1 to interferon response and pigmentation defects. Therefore, I feel that information on how Adar1 exactly controls these processes in neural crest would be important. The authors nicely show that the Schwann cell and pigmentation phenotype of Adar1 inactivation can be rescued by *Ifih1* inactivation. This allows them to conclude that the observed defects "result from a mechanism that is MDA5 dependent", but we have no idea about the nature of this mechanism. The authors refer to previous literature ("MDA5 is ...triggering innate immune system activation in other tissues"; rebuttal letter), implying that innate immune system activation causes the differentiation defect in Schwann cells and apoptosis in melanocytes. The culture experiments (obviously done in the absence of innate immune cells) rather speak against this hypothesis. Are innate immune cells activated and recruited at all to the respective sites at the early time point at which the defects arise (prenatally for the melanocyte lineage)?

As suggested, the authors have now investigated further neural crest derivatives in Adar1 cko animals and found no defects in the ENS, heart, and craniofacial structures. The analysis (Suppl Fig. 5) is a bit superficial (macroscopic, histopathological). Note that by such an analysis, the nerve phenotype likely would have been missed as well.

Furthermore, as stated before, I find it intriguing that Adar1 inactivation in neural crest cells and the consequent increase in ISG expression appears to affect different neural crest derivatives in substantially different manners (survival vs differentiation vs no response). The authors come up

with a possible (at this point plausible but hypothetical) explanation for why Schwann cells survive Adar cko, but it remains unclear how/whether A to I RNA editing, ISG expression changes, and innate immune cell activation affect late differentiation steps. On the same token, why would other derivatives be fully protected? Don't they express the receptors mediating immune recognition?

Moreover, with respect to the examination of additional neural crest derivatives, the fold change in ISG expression upon cko Adar1 is quite prominent despite analysis of the entire tissue (ie gut rather than ENS, heart rather than the outflow tract harboring neural crest-derived cells). Could it be that the ISG expression response is systemic in the cko mice and not intrinsic to/specific for neural crest derivatives? Related to this, in Fig 5, are melanocytes specifically affected or are all neural crest cells (and maybe even neural tube cells) in the explant decreased in numbers?

Reviewer #1 (Remarks to the Author): *The revised manuscript from Gacem et al. on the role of Adar1 in melanocyte survival and Schwann cell myelination in mouse embryonic development has considerably improved. One of the main caveats was that the data, although potentially robust, were not fully explored and described. This concern has been addressed with the rigorous analysis of the existing data and the addition of convincing quantifications. The Schwann cell myelination phenotype is strengthened as the authors were able to rule out the possibility of an axonal defect or a decrease in the number of Schwann cells. The conclusions presented in the manuscript that were hypothetical are now suitably supported by the results and quantifications.*

Finally, the authors have nicely completed the story by pushing the melanocyte analysis further. New in vitro experiments rescuing melanocyte function have been performed and were added to the manuscript along with an expanded discussion clarifying the pigmentation anomalies observed in the Adar1 mutants.

We thank the reviewer for comments that obviously helped improve the paper.

Reviewer #2 (Remarks to the Author): *The authors have addressed the comments raised during review. Given the ultimate conclusion that this is an editing dependent, Mda5 dependent phenotype the reference that best demonstrates this is PMID: 28874170 rather than relying on the unpublished perinatal data/observations from the Stetson lab.*

We added the reference mentioned by the reviewer. Adar1^{E861A/E861}; Ifih1^{-/-} animals presented in the supplementary videos indeed show no pigmentation defects. However, because we used the Adar1^{-/-} allele, we believe both this reference and the unpublished results of D. Stetson are of importance to support our analysis.

Reviewer #3 (Remarks to the Author): *The authors performed new experiments and provide further data to strengthen their study. However, I still feel that more mechanistic insights into how Adar1 controls neural crest development would be required for publication in this journal.*

1) As mentioned before, others have previously linked Adar1 to interferon response and pigmentation defects. Therefore, I feel that information on how Adar1 exactly controls these processes in neural crest would be important. The authors nicely show that the Schwann cell and pigmentation phenotype of Adar1 inactivation can be rescued by Ifih1 inactivation. This allows them to conclude that the observed defects “result from a mechanism that is MDA5 dependent”, but we have no idea about the nature of this mechanism. The authors refer to previous literature (“MDA5 is ...triggering innate immune system activation in other tissues”; rebuttal letter), implying that innate immune system activation causes the differentiation defect in Schwann cells and apoptosis in melanocytes. The culture experiments (obviously done in the absence of innate immune cells) rather speak against this hypothesis. Are innate immune cells activated and recruited at all to the respective sites at the early time point at which the defects arise (prenatally for the melanocyte lineage)?

As already mentioned in the results section page 9, analysis of electronic microscopy pictures revealed “There was no sign of [...] B- or T-cell or macrophage invasion in the mutants compared to controls sciatic nerves at P4”.

To complement this observation, we performed immunohistochemistry experiments using anti-CD45 and a cocktail of F4/80+CD11b+CD68 antibodies to detect all hematopoietic cells. The

immunohistochemistry experiments were performed on controls and mutants' skin sections at E18.5 and P4 and on sciatic nerves sections at P4.

In skin, both stains were found similar in controls and mutants. Quantifications of the number of CD45 positive cells confirmed these observations, revealing no significant differences between the two genotypes.

In sciatic nerves, the percentage of CD45 positive cells was slightly increased (4%) in mutants compared to controls, but that of F4/80+CD11b+CD68 cells was unchanged. Of note, the slight increase was observed in only two out of the four mutants analyzed, although they all present with reduced myelin. While we cannot exclude that immune cells are recruited later on (but the early death of mutants precludes analysis), these results suggest that innate immune cells recruitment is rather limited at the site of alterations up to P4, and likely does not account for the observed phenotypes.

These observations and our *in vitro* results (indeed *done in the absence of innate immune cells*) are compatible with results published by Masaki et al. who showed that Schwann cells, which are typically nonimmune cells, possess the capacity to elicit innate immune response by expressing a large number of innate immune genes in response to *Mycobacterium leprae* (ML) infection (Masaki, 2012). This also stands for melanocytes (see Gasque, 2015, showing that emerging innate immune activities of human melanin-producing cells can sense and respond to bacterial and viral infections).

To answer reviewer concern and clarify this point, we:

1. Present CD45 staining and quantification on skin sections at E18.5 and P4 in new supplementary Figure 2 and both antibodies staining along with quantifications on sciatic nerves sections at P4 in new supplementary Figure 3. Consequently, all supplementary figures were renamed (Supp Fig3 now Supp Fig4 etc.....).
2. Mention within the discussion section that Schwann cells and melanin-producing cells, which are typically nonimmune cells, possess the capacity to elicit innate immune response along with adequate references.
3. Changed the formulation "*MDA5 is ...triggering innate immune system activation*" in the whole manuscript, as it was inappropriate.

2) As suggested, the authors have now investigated further neural crest derivatives in Adar1 cko animals and found no defects in the ENS, heart, and craniofacial structures. The analysis (Suppl Fig. 5) is a bit superficial (macroscopic, histopathological). Note that by such an analysis, the nerve phenotype likely would have been missed as well. Furthermore, as stated before, I find it intriguing that Adar1 inactivation in neural crest cells and the consequent increase in ISG expression appears to affect different neural crest derivatives in substantially different manners (survival vs differentiation vs no response). The authors come up with a possible (at this point plausible but hypothetical) explanation for why Schwann cells survive Adar cko, but it remains unclear how/whether A to I RNA editing, ISG expression changes, and innate immune cell activation affect late differentiation steps. On the same token, why would other derivatives be fully protected? Don't they express the receptors mediating immune recognition?

As stated by the reviewer, increased expression of ISGs due to A-to-I editing alterations appears to affect neural crest derivatives in substantially different manners. Some of the ISGs might trigger

apoptosis of melanocytes as shown in the case of infectious diseases and skin inflammation (see Gasque, 2003).

We propose that activation of ISGs might also lead to the differentiation defects observed in Schwann cells. As mentioned in previous reviewer 2 comment, it is however difficult to ascertain the causality of such a large number of genes (2253 are activated in sciatic nerves of mutants compared to controls), and identify primary versus secondary events. Nevertheless, we observed that some of them are early NC-cell genes or negative regulators of myelination, and part of the injured response we identified. As discussed page 18-19, we propose that aberrant expression/maintenance of some of these (especially if they are also involved in early phases of repair process), may dysregulate the balance between positive and negative regulators of myelination, explaining the block of myelination observed (see Supplementary table 2). Of note, *Mycobacterium leprae* infection of Schwann cells was also shown to turn off differentiation/myelination-associated genes and reactivate developmental-associated genes/transcription, changing cell fate through an unknown mechanism (Masaki, 2012).

The reason why no (or incompletely penetrant) alterations were observed in heart, gut or craniofacial structures, could at least partially be due to restrictive expression/functions of ISGs. Indeed, ISGs are expected to impact NC development /differentiation only if they are expressed during specific time windows, alone or in combination with other genes controlling normal development of tissue of interest (for example upregulation of known negative regulators of myelination such as *EGR1* could impact myelination, but have little impact on other NC derivatives). As suggested by the reviewer, subtle defects affecting heart, gut or craniofacial structures could also have been missed. However, our observations seems to fit very well with phenotypes observed in patients. Pigmentation defects have indeed been described in patients presenting with *ADARI* mutations, and once within a patient with peripheral anomalies (see discussion page 20). However, no craniofacial or NC-related heart alterations have been reported.

As suggested by the reviewer, expression level of the receptors mediating immune recognition, *Mda5* (*Ifih1*) in particular, might also be important. To test this possibility, we analyzed its relative expression level in skin, mandible, heart, and gut compared to sciatic nerves at E18.5 and P4 by RTqPCR. No significant difference was evidenced (see table underneath).

relative expression level of Ifih1 (tissue of interest compared to sciatic nerves)				
	skin/sciatic nerves	mandible/sciatic nerves	heart/sciatic nerves	gut /sciatic nerves
E18.5	1.12±0.12	0.69±0.10	2.75±0.26	2.73±0.26
P4	1.56±0.36	0.53±0.10	1.25±0.12	1.16±0.30

We finally performed CD45 staining on guts and hearts of mutants and controls, No difference was detected between the two genotypes, suggesting absence of immune cells recruitment. Analysis of the level of expression of other receptors mediating immune recognition therefore seems irrelevant.

To answer reviewer concerns we:

1. Modified paragraph page 18-19 of the discussion to clarify mechanism that could lead to differentiation defects.

You can now read “Nevertheless, some of **these ISGs** are early NC-cell genes or negative regulators of myelination, and/or are part of the injured response described above, suggesting that **chronic activation of some of them may dysregulate the balance between positive and negative regulators of myelination,**

explaining the observed blockade of myelination. **Of interest**, 61 genes known to be transiently expressed after injury (at one and five days post injury but not later) are abnormally maintained in mutants with a fold-change > 5, including *Soat2*, *Vgf*, *Egr1*, *Ch25h* who have known functions in myelination/demyelination processes (see supplementary table 2, genes indicated in red). Their deregulation could therefore participate to the phenotype genesis.

A column was also added to Supplementary Table 2 to mention if genes are known ISGs; the 61 genes are indicated in red.

2. Tempered our conclusions on other NC derivatives. We now mention that we did not see major alterations at the stages analyzed and with the techniques used, but that more subtle alterations could have been missed.

3. Mention *Ifih1* expression level in different tissues in the discussion section page 20.

3) Moreover, with respect to the examination of additional neural crest derivatives, the fold change in ISG expression upon cko Adar1 is quite prominent despite analysis of the entire tissue (ie gut rather than ENS, heart rather than the outflow tract harboring neural crest-derived cells). Could it be that the ISG expression response is systemic in the cko mice and not intrinsic to/specific for neural crest derivatives?

To determine if ISGs expression is systemic, we collected livers from 3 mutants and 7 controls at P4 to test the ISGs signature by RTqPCR. We chose this organ because its function as a major immune organ is well appreciated (and contains no NC cells).

No significant upregulation of the ISGs signature was observed in mutants compared to control littermates. The ISGs response is therefore not systemic but intrinsic to/specific for NC derivatives.

To answer reviewer concern we present this information in the results section page 11, in new supplementary Figure 8 and accompanying legend, and in the discussion page 18.

Related to this, in Fig 5, are melanocytes specifically affected or are all neural crest cells (and maybe even neural tube cells) in the explant decreased in numbers?

The medium we used is known to mainly favor survival and growth of melanocytes, precluding analysis of other NC derivatives requested by the reviewer without bias.

However, we would like to mention that no difference was observed between explants cultures of controls and mutants during the first 7 days of culture. During this time window, cells were clearly visible outside of the explant of both controls and mutants and are just starting to differentiate into melanocytes. Culture for a period for an extra 14 days then favored expansion of melanocytes in controls but not in mutants (results presented in Figure 7). As shown in the pictures at low magnification presented below, explants and other cells present in the culture did not seem to be drastically altered at the end of the culture (control on the left and mutant on the right).

To answer reviewer concern without lengthening the paper, we slightly modified the legend of the Figure 7 to mention this information.

REVIEWERS' COMMENTS:

Reviewer #3 (Remarks to the Author):

In this re-revised version of their manuscript, the authors have addressed all my points by further experiments and adjustments in the text.

In particular, they found that the innate immune system is apparently not involved in development of the reported phenotype, in contrast to what the authors have previously suggested. Accordingly, the authors have now changed the statement "MDA5 is ...triggering innate immune system activation" throughout the entire manuscript, as it was inappropriate.

Moreover, they now provide evidence that the phenotypic differences in neural crest-derived structures are neither due to differential ISG responses nor to differential expression of receptors mediating immune recognition.

I feel that these new data are important, although -given their negative nature- they obviously don't really help to provide more mechanistic insights into how Adar1 controls neural crest development. However, the discussion regarding this issue has been clearly improved.

REVIEWERS' COMMENTS:

Reviewer #3 (Remarks to the Author): In this re-revised version of their manuscript, the authors have addressed all my points by further experiments and adjustments in the text. In particular, they found that the innate immune system is apparently not involved in development of the reported phenotype, in contrast to what the authors have previously suggested. Accordingly, the authors have now changed the statement "MDA5 is ...triggering innate immune system activation" throughout the entire manuscript, as it was inappropriate. Moreover, they now provide evidence that the phenotypic differences in neural crest-derived structures are neither due to differential ISG responses nor to differential expression of receptors mediating immune recognition.

I feel that these new data are important, although -given their negative nature- they obviously don't really help to provide more mechanistic insights into how Adar1 controls neural crest development. However, the discussion regarding this issue has been clearly improved.

We thank the reviewer for comments that obviously helped improve the paper.